# Key drivers of fertility levels and differentials in India, at the national, state and population subgroup levels, 2015–2016: An application of Bongaarts' proximate determinants model

**Susheela Singh**[1]*, **Chander Shekhar**[2], **Akinrinola Bankole**[1], **Rajib Acharya**[3], **Suzette Audam**[1], **Temitope Akinade**[1]

**1** Guttmacher Institute, New York, New York, United States of America, **2** Department of Fertility Studies, International Institute for Population Sciences (IIPS), Mumbai, India, **3** Population Council, New Delhi, India

* ssingh@guttmacher.org

## Abstract

### Objectives

The transition to small family size is at an advanced phase in India, with a national TFR of 2.2 in 2015–16. This paper examines the roles of four key determinants of fertility—marriage, contraception, abortion and postpartum infecundability—for India, all 29 states and population subgroups.

### Methods

Data from the most recent available national survey, the National Family Health Survey, conducted in 2015–16, were used. The Bongaarts proximate determinants model was used to quantify the roles of the four key factors that largely determine fertility. Methodological contributions of this analysis are: adaptations of the model to the Indian context; measurement of the role of abortion; and provision of estimates for sub-groups nationally and by state: age, education, residence, wealth status and caste.

### Results

Nationally, marriage is the most important determinant of the reduction in fertility from the biological maximum, contributing 36%, followed by contraception and abortion, contributing 24% and 23% respectively, and post-partum infecundability contributed 16%. This national pattern of contributions characterizes most states and subgroups. Abortion makes a larger contribution than contraception among young women and better educated women. Findings suggest that sterility and infertility play a greater than average role in Southern states; marriage practices in some Northeastern states; and male migration for less-educated women. The absence of stronger relationships between the key proximate fertility determinants and geography or socio-economic status suggests that as family size declined, the role of these determinants is increasingly homogenous.

**Data Availability Statement:** The primary data used in this study is from the India National Family

Health Survey (NFHS-4) conducted in 2015-16. The data from this survey are available for download through the Demographic and Health Surveys (DHS) Program (https://www.dhsprogram.com/).

**Funding:** This article is based on research funded by the Bill & Melinda Gates Foundation, grant INV-002939 (https://www.gatesfoundation.org/) and the David and Lucile Packard Foundation, grant 2019-69110 (https://www.packard.org/). The funders had no role in study design, data collection and analysis, decision to publish, or preparation of the manuscript.

**Competing interests:** The authors have declared that no competing interests exist.

## Conclusions

Findings argue for improvements across all states and subgroups, in provision of contraceptive care and safe abortion services, given the importance of these mechanisms for implementing fertility preferences. In-depth studies are needed to identify policy and program needs that depend on the barriers and vulnerabilities that exist in specific areas and population groups.

## Introduction

India is the second most populous country in the world and houses nearly a fifth of the world's population. According to its National Population Policy, India has set itself the goal of achieving population stabilization by 2045 [1]. While various policy and program interventions have been made from time to time to address sexual and reproductive health care needs, evidence on the key drivers of average family size, including the relative importance of these drivers, and how they vary across population subgroups, may further inform policies and programs. This paper examines key drivers of fertility for India, using the most recent available survey data, the National Family Health Survey (NFHS-4), conducted in 2015–16 [2]. It focuses on the four proximate determinants that can explain about 96% of reduction in fertility from the potential biological maximum—marriage, contraceptive use, induced abortion and the duration of postpartum infecundability [3]. The relative importance of these factors as determinants of family size were assessed by applying the proximate determinants model of fertility, developed in the 1950s [4], and operationalized and improved over the past four decades [5–8].

This work builds on a recent report [9] that looked at trends in the role of the proximate determinants among all women of reproductive age by state over the period 1998–2016, using three rounds of the National Family Health Surveys and covering the whole country and the largest 18 states. An important finding of that study is that while increased use of contraception was a major factor underlying the decline in family size between 1998/99 and 2005/06, it was replaced by two key factors that explained the further decline in fertility between 2005/06 and 2015/16: marriage and abortion. The contribution of postpartum infecundability gradually declined over this period. That study was only able to incorporate abortion at the national level given the data available at the time.

This study expands on previous studies in several important areas: We cover all 29 states in the country, with analyses of the NFHS-4, conducted in 2015–2016 and for the first time using a large enough sample size to provide estimates for all 29 states; we also for the first time provide estimates of the impact of the four key determinants for key demographic and socioeconomic population subgroups, namely age, place of residence, education, household wealth status and caste. In addition, we incorporate abortion into the model at all levels, national, for each state and for population subgroups—drawing on a recent study that provides estimates for the same time period as the NFHS-4, and that provides the basis for developing state and subgroup estimates (Singh et. al. 2018). Finally, we introduced two modifications to the model that were important in the Indian context: hysterectomy for non-contraceptive reasons is practiced at a notable level (reported by 4% of married women 15–49 in 2015–16), and this factor is incorporated into the estimate of the impact of postpartum infecundability and into the estimate of the impact of contraception, through its overlap with contraceptive use.

The objective of this paper is to provide comprehensive and detailed estimates of the four principal determinants of fertility level—nationally, by state and for key demographic and social-economic population subgroups—using the most recent data available, and with adjustments to the model to take the context of India into account. The goal is to inform policies and programs that address gaps in access to sexual and reproductive health (SRH) services at the national, state and subgroup level, through providing evidence on the role of the key determinants of fertility level across states and population groups.

It is important to analyze differences across states and population subgroups in India because of the wide variation across the country in terms of access to education, wealth differentials and social disenfranchisement (captured by 'caste'), all factors that influence sexual and reproductive behaviors including childbearing. For example, in 2015–16 the Total Fertility Rate (TFR) varied from 1.17 in Sikkim to 3.41 in Bihar; the mortality rate for children varied from 7 deaths per 1000 children under five in Kerala to a rate of 78 in Uttar Pradesh; and use of modern contraceptive methods varied from 13 percent in Manipur to 69 percent in Andhra Pradesh [2]. As of 2015–16, 19 states had attained below replacement level fertility (TFR < 2.1) while six states in the central, east, and north-eastern parts of the country reported TFRs of 2.5 or higher. Furthermore, differentials in these measures by population characteristics are also considerably large (e.g., nationally, the TFR among women with no literacy was 3.06 versus 1.71 among women with 12 or more years of schooling), although these differentials are beginning to narrow in many states.

There has been substantial progress in access to contraceptive services in India since 1952, when India became one of the first countries in the world to initiate a state-sponsored family planning program, [10, 11]. The program was estimated to have averted 168 million births by the year 1996 [12]. In recent times, use of modern contraceptives has risen from 37 percent during 1992–93 [13] to 48 percent during 2015–16 [2]. However, there are several individual, social, and systemic barriers that Indian couples encounter when accessing and using family planning services; in addition, health concerns, side effects and other method-related reasons are also important barriers to use of modern contraception. Two useful summary indicators of barriers to using modern contraception are unmet need for modern contraceptive methods—the proportion of women who are fecund, do not want a child soon but are not using a modern method—and the proportion of total demand satisfied by modern methods. In 2015–16, 19% of married women aged 15 to 49 had an unmet need for modern methods of contraception nationally, ranging from 5% in Andhra Pradesh to 41% in Manipur [2]. Over 8 out of 10 women (80.6%) had their contraceptive demand satisfied by modern methods in India, lowest in Manipur (24%) and highest in Andhra Pradesh (94%). Unmet need was particularly high among married women aged 15–24 (27 percent), those with lower parity (24 percent among those with zero or one child), and poor women (23 percent among the lowest wealth quintile). A recent study in six states found that roughly half of pregnancies in 2015 were unintended, and about two-thirds of these unintended pregnancies ended in induced abortion [14].

India legalized abortion under broad criteria in 1971 [15] and in 2002, an amendment to the 1971 Act was passed, approving medication abortion (MA) [16]. A further amendment was passed in 2021, expanding access to later term abortions and improving access for many vulnerable subgroups, including unmarried women. Nevertheless, access to and utilization of safe and legal abortion services is still limited. While three-fourths of the Indian population live in rural areas, abortion services are rarely available at rural health facilities because physicians are not available to staff them [17–19] and other health professionals are not approved to provide abortion. Available safe abortion services are underutilized due to numerous individual and community-level factors, such as lack of awareness of the legality of abortion, limited understanding of the implications of unsafe abortion and lack of information on availability of

safe providers and methods [20]. Representative estimates of abortion incidence are available for six states and at the national level for 2015, coinciding with the timing of data collection for the NFHS-4. The national abortion rate was 47 per 1000 women ages 15–49, and the abortion rate ranged widely among the six states from 37 for Tamil Nadu to a rate of 70 in Assam [14]. The study also permits approximate estimates for all states based on the average rates for six major regions [14].

The age at marriage for women in India increased over the last decade–the sharpest ever increase in the country's history. The proportion of women marrying before 18 among women age 20–24 declined between 2005–06 and 2015–16 by 20 percentage points nationally–about 2 percentage points per year [2, 21]. Wide state-level variations were observed in early marriage: the proportions of women aged 18–29 years who married before age 18 were highest in Bihar and West Bengal (42% and 44% respectively) and lowest (9%-10%) in four states (Himachal Pradesh, Jammu & Kashmir, Kerala and Punjab) [2]. The median duration of post-partum insusceptibility (the average number of months women are either amenorrheic or abstaining from intercourse after giving birth) has been on the decline too–from 8.1 months in 2005–06 to 6.6 months in 2015–16 [2, 21]. NFHS 2015–16 data showed that median postpartum insusceptibility varied widely by place of residence (5.9 for urban residents versus 7.0 for women living in rural areas), education (5.8 among those having 12 years or more of schooling versus 7.6 among those who did not go to school) and wealth (5.1 among women from the highest wealth quintile versus 8.2 among women from the lowest wealth quintile) [2].

## Materials and methods

### Data source

This paper analyzes data from the fourth round of the India National Family Health Survey (NFHS-4) conducted in 2015–16. The survey collected information on standard Demographic and Health Survey indicators, including the characteristics of households and respondents, maternal and child health (including breastfeeding), marriage, fertility and contraceptive use. The NFHS-4, which covers all 29 states and 7 union territories, is a nationally representative cross-sectional survey that employed a two-stage stratified probability proportional to size sampling design. In the first stage, primary sampling units (PSUs) were selected—villages in rural areas and Census Enumeration Blocks (CEBs) in urban areas. In the second stage, 22 households were selected using systematic sampling from each PSU (see details, page 1, IIPS and ICF, 2017). All women of reproductive age (15–49 years) living in each selected household were interviewed. Data were collected from 699,686 women aged 15–49 living in 601,509 households, who comprise the analytical sample for this article. Details of the survey and sampling procedure have been published elsewhere [2]. A 2015 study on abortion incidence that provides estimates of national and regional abortion rates and rates for six states (Assam, Bihar, Gujarat, Madhya Pradesh, Uttar Pradesh and Tamil Nadu) is the data source for calculating the contribution of abortion; details on the study design and sampling are available [14].

### Analytical approach

To assess the relative role of the four main factors that influence the level of fertility—marriage, contraceptive use, induced abortion and postpartum infecundability—we apply the Bongaarts Proximate Determinants (PD) model to the 2015–16 NFHS data, nationally, for each state and for selected population subgroups nationally and within each state. We analyze differences among subgroups for the following characteristics: age, residence, years of schooling, wealth status and caste. For the national and state level analyses we used five-year age groups: 15–19, 20–24, 25–29, 30–34, 35–39, 40–44 and 45–49. Residence is defined as a two-category variable,

urban and rural. Women's years of schooling is classified into three categories: < 5 years, 5–9 years and 10+ years; we combined women with no education (illiterate) and those with 1–4 years of schooling into one category because for some states, the size of the former or the latter group is too small to stand as a separate category. Furthermore, there is little substantive difference between the two groups and in India, for policy purposes the government uses the cutoff of <5 and 5 or more years of schooling as indicators of educational attainment. We classified the household wealth index into tertiles–poor (lowest one-third), middle class (middle one-third) and rich (highest one-third) [22]. Finally, caste was categorized into three groups, from low to high social status, using labels applied in India: scheduled castes and tribes (SCs/STs), other backward class (OBC) and Others (which includes all castes and groups other than those belonging to the two lower status categories). For calculating indices of the four proximate determinants at the subgroup level, age was classified as a three-category variable to take into account small sample size: 15–24, 25–34 and 35–49. We applied a cutoff of 50 unweighted cases below which results are not presented. Selected key indicators that are used for the construction of the indices are presented in S1 Table.

As noted in the Introduction, hysterectomy occurs with some frequency in India and some of these procedures are not for contraceptive purposes. For example, in our sample, nationally, 4 percent of married women had a non-contraceptive hysterectomy, and the proportion was as high as 11 percent in Bihar and Goa. Therefore, it is likely that non-contraceptive hysterectomy contributes to infecundability. As a result, our measure of infecundability takes into account the roles of full breastfeeding, amenorrhea and postpartum abstinence (for women who had a birth in the three years prior to the date of interview) as well as non-contraceptive hysterectomy (also in the three years prior to the date of interview).

## Analysis

**Calculation of the four proximate determinants and other model parameters.**   The impact of the four proximate determinants (marriage, contraception, induced abortion and postpartum infecundability) was examined using the Bongaarts Proximate Determinants (PD) model. The multiplicative PD model, defining the total fertility rate as the product of four indices and total fecundity (*TF*) [5, 6], can be expressed as:

$$TFR = C_m \times C_c \times C_a \times C_i \times TF$$

Where,

*TFR* is total fertility rate

$C_m$ is the index of marriage

$C_c$ is the index of contraception,

$C_a$ is the index of induced abortion,

$C_i$ is the index of postpartum infecundability, and

*TF* is total fecundity

*Index of marriage.* The index of marriage ($C_m$) is calculated based on the proportion currently married by five-year age groups among women of reproductive age, which is a proxy for sexual activity and likelihood of pregnancy. The value of the index ranges from 0 to 1; when all women of reproductive age in the population are married at age 15 and remain married throughout the reproductive years (15–49), the index equals 1, and when no woman is married, the index equals 0.

The age-specific index of marriage is estimated as:

$$C_m(a) = m(a)$$

Where, $m(a)$ is the proportion currently married in age group '$a$.'

The aggregate index of marriage is:

$$\mathbf{C_m} = \sum C_m(\mathbf{a}) \times w_m(\mathbf{a})$$

Where,

the weight, $w_m(\mathbf{a}) = \dfrac{f_m(\mathbf{a})}{\sum f_m(\mathbf{a})}$

The age-specific marital fertility rate, $f_m(\mathbf{a})$, is calculated by dividing the births of married women of age (**a**) by the number of married women in the same age. Because of the low proportion married among women aged 15–19, (**a**) for this group may be subject to large random errors. To guard against this, the marital fertility rate for women aged 15–19 is assumed to be three-quarters of the marital fertility rate for women aged 20–24 in line with previous studies [23, 24].

*Index of contraception.* The index of contraception ($C_c$) measures the inhibiting effect of contraceptive use on fertility in the population. The index is estimated as 1 minus the proportion of fecund women who are effectively protected from becoming pregnant through use of contraception. We defined the index for currently married women who are considered sexually active. The index $C_c$ equals 1 when no contraception is used in the population and 0 when all fecund married women use methods that are 100% effective.

The age-specific index of contraception is expressed as:

$$C_c(a) = 1 - r(a) \times (u(a) - o(a)) \times e(a)$$

Where,

$r(a)$ = fecundity adjustment at age $a$

$u(a)$ = contraceptive prevalence among married women of reproductive age $a$

$o(a)$ = overlap of contraceptive use with infecundability at age $a$

$e(a)$ = average effectiveness of contraception at age $a$

The aggregate index of contraception is calculated as:

$$C_c = \sum C_c(a) \times w_c(a)$$

Where,

The weight $w_c(a) = \dfrac{f_n(a)}{\sum f_n(a)}$,

$f_n(a)$ = the marital fertility rate in the absence of any deliberate efforts to prevent pregnancy or birth at age '$a$'

The fecundity adjustment, $r(a)$, which adjusts contraceptive use prevalence among all exposed (that is, married and fecund) women to account for the fact that method use is higher among fecund women than among infecund women, varies by age group. We use the values from a recent analysis of data from 36 countries: For each age group 15–19 through 45–49, are 0.62, 0.81, 0.99, 1.08, 1.14, 1.26 and 1.62, respectively [6]. Contraceptive prevalence ($u(a)$) is the proportion of married women at age ($a$) using any method of family planning.

To measure overlap between contraceptive use and infecundability, we constructed a variable with a value of 1 or 0. The value is 1 if a married woman was using a method of contraception at the time of the survey and she was also fully breastfeeding or amenorrheic or abstaining from sex during the postpartum period or if she had undergone a hysterectomy not for contraceptive purposes in the past three years. The proportion of married women with a value of 1 gives the extent of overlap between contraception and infecundability in the population.

Average contraceptive use effectiveness at a given age is defined as:

$$e = \sum e(m) \times \frac{u(m)}{u}$$

Where,

$u$ = Proportion of married women using any contraception

$e(m)$ = Method specific use effectiveness

$u(m)$ = Proportion of married women using a specific method

The proportion of married women using contraception and the proportion using specific methods are obtained from the NFHS-4. The method specific use effectiveness is a constant for each method and is the complement of the method specific use failure rates calculated using global sources [25].

*The index of induced abortion.* The index of induced abortion ($C_a$) is a function of the number of births averted by an abortion. The number of births averted per induced abortion (b) is assumed to be largely independent of the age of woman, but strongly related to the practice of contraception following the induced abortion [5].

i. The age-specific formula for the index is:

$$C_a(a) = \frac{f(a)}{(f(a) + bab(a))}$$

Where,

$f(a)$ = fertility rates for age '$a$'

$ab(a)$ = abortion rate for age '$a$'

$b = \frac{14}{(18.5 + i(a))}$, and.

$i(a)$ = average duration of infecundability for age '$a$'

ii. The aggregate index of induced abortion is expressed as:

$$C_a = \sum C_a(a)w_a(a) \approx \frac{TFR}{TFR + bTAR}$$

Where,

$TAR$ = Total Abortion Rate

Following Bongaarts [6], we assumed that $C(a) = \sum C_a(a)$, therefore, the aggregate index of induced abortion was not calculated from the age-specific index. Rather we used the aggregate formula. The national and regional abortion rates and rates for six states (Assam, Bihar, Gujarat, Madhya Pradesh, Uttar Pradesh and Tamil Nadu) were obtained from published findings from a study of the incidence of induced abortion in India in 2015 [14]. To obtain subgroup specific abortion rates needed to calculate both the index of abortion and the aggregate index of contraception, we applied Jayachandran and Stover's [9] assumption that abortion behavior

follows a similar pattern as unmet need for contraception by age for state-level or subgroup populations. Based on this assumption and using the distribution of women with unmet need from the NFHS-4 for each subgroup and the number of abortions at the national, regional or state level, we estimated the numbers and rates of abortion for each subgroup category. Because this estimation resulted in age-specific abortion rates that were very high at the oldest age groups for some states, we adjusted the state age-specific rates. We calculated the final state age-specific abortion rates as the national age-specific rates multiplied by the ratio of the overall national abortion rate to the overall abortion rate for the state. The assumption that the pattern of age-specific abortion behavior among women is similar to that of unmet need for modern methods does not work as well at the oldest age groups, likely because of high incidence of infecundity at the upper end of the reproductive age-range. Consequently, the abortion rates produced by this assumption at those ages tend to be too high, prompting the decision to make this adjustment.

*Index of infecundability*. In our analysis, the index of infecundability ($C_i$) measures the inhibiting effect of four factors—breastfeeding, amenorrhea, postpartum abstinence, and non-contraceptive hysterectomy—on fertility in the population. To calculate the average duration of infecundability, we selected the factor among these four that had the longest duration for each woman. We then summed up the durations for women in each age group and divided the result by the number of women in that age group to obtain the average duration of infecundability for that age group.

The age-specific version of the index is expressed as:

$$C_i(a) = \frac{20}{(18.5 + i(a))}$$

Where,

$i(a)$ = average duration of infecundability at age $a$.

The aggregate index is in turn expressed as:

$$C_i = \sum C_i(a)w_i(a) \approx C_i = \frac{20}{(18.5 + i)}$$

Where

$i$ = average duration of infecundability.

In line with Bongaarts [6], we assumed that $C_i(a)$ varies little by age, therefore, we calculated the aggregate index of infecundability using the aggregate formula, rather than deriving it from the age-specific index.

**Predicted ASFR, TFR and contribution of the four proximate determinants to fertility.** After estimating the indices for the four principal proximate determinants (marriage, contraceptive use, induced abortion and postpartum infecundability) at the national and state levels, and for four subgroups (residence, education, caste and wealth status), we used the values of the indices to determine other measures.

*Predicted TFR and ASFR*. The predicted total fertility rate (TFRe) and the predicted age-specific fertility rate (ASFRe) can be obtained as the product of the four indices and the fecundity rate (i.e. the biological maximum average number of births per woman which assumes that all women marry at age 15 and all married women make no voluntary efforts to control fertility). For this analysis, at the national, state and subgroup levels, we used the national total fecundity (*TF*) of 10.9 per woman and the age-specific fecundity rates estimated by Jayachandran and Stover [9] from the NFHS-3, a source that is external to the source of the data we analyzed (NFHS-4), consistent with recommendations for applying the PD model. These age-specific

fecundity rates (expressed as the number of births per woman per year) are: 0.484 for age 15–19, 0.540 for age 20–24, 0.531 for age 25–29, 0.448 for age 30–34, 0.290 for age 35–39, 0.167 for age 40–44 and 0.109 for age 45–49.

*Proportion of the reduction in fertility from the biological maximum of the total fecundity that is due to each proximate determinant.* The fertility-inhibiting effect of each of the four proximate determinants can be estimated as the complement of its index. For instance, suppose the index of contraception is 0.56, the inhibiting effect of contraceptive use will be 1–0.56 = 0.44. This means that 44 percent of the reduction in fertility from the Total Fecundity rate to marital total fertility rate is due to contraceptive use. Assuming that the four proximate determinants independently and completely account for the reduction in fertility from total fecundity to the observed TFR, we can estimate the relative contribution of each determinant. The contribution of each determinant to fertility reduction is estimated as the ratio of the natural logarithm of the index for that determinant to the sum of the natural logarithm of the four indices, multiplied by 100. For instance, the percentage contribution of contraception to reduction in fertility from total fecundity to total fertility rate will be $\log_e(C_c) * \frac{100}{(\log_e(C_m) + \log_e(C_c) + \log_e(C_a) + \log_e(Ci))}$. The contributions of the four indices will then sum up to 100 percent.

The role of each of the four proximate determinants—marriage, contraception, induced abortion and postpartum infecundability—in determining the reduction in fertility from the Total Fecundity (*TF*) to the Total Fertility Rate (TFR)—is evaluated using the indices and the percent of the reduction that is attributed to each of the four factors. The model also permits calculation of an estimated TFR: The difference between the estimated and actual TFRs, referred to as the "residual," indicates the contribution of unmeasured factors (not attributable to the four that are measured) to the actual TFR and the quality of data and assumptions used to calculate the four determinants. The model provides valuable estimates of the relative importance of the four key determinants, and differences and similarities across states and population subgroups.

## Results

We present results at the national level and for each of the 29 states. Union Territories are included in national estimates, but results are not presented for them separately because of small sample size. Data are presented grouping the 29 states by major region to assess the extent to which the patterns in the contribution of the four determinants are region-specific, given similarities in sociocultural contexts within regions. We also analyzed patterns with states grouped according to level of the TFR (S5 Table), but no specific patterns emerged with respect to proportionate contribution of the determinants, possibly because by 2015–16 the TFR ranged narrowly across states. In assessing the extent to which state estimates are different from national estimates we deem a difference as significant if a determinant's proportionate contribution to fertility reduction is more than five percentage points different from the contribution of that determinant for comparable subgroups at the national level, or by comparison with other subgroups with similar socio-economic characteristics within states. This approach was chosen under the assumption that if the four determinants contribute equally to the reduction in *TF* (i.e., 25 percent each) then a 5 percentage points change in any determinant's contribution, i.e. (5/25*100) is equal to 20 percent, which can be considered as a significant change in the proportionate contribution.

### Impact of proximate determinants at the national level

At the national level, the values of the indices and the percent contribution of the four determinants show that marriage makes the largest relative contribution to the TFR: It has the smallest index (0.59) and accounts for 36% of the reduction from Total Fecundity (*TF*) (Table 1).

**Table 1. Estimates of the indices of the four key proximate determinants on fertility, actual TFR, estimated TFR, residual, and estimates of percent contribution of each proximate determinant to fertility reduction: National and by state, India 2015–16.**

| National/ Region State | Indices | | | | Fertility | | | % contribution of determinant to fertility reduction from total fecundity§ | | | | Number of Women: Unweighted Ns |
|---|---|---|---|---|---|---|---|---|---|---|---|---|
| | Marriage (Cm) | Contraception (Cc) | Induced Abortion (Ca) | Postpartum Infecundability (Ci) | TFRa* | TFRe** | Residual† | Marriage | Contraception | Abortion | Postpartum Infecundability | |
| **National** | 0.59 | 0.70 | 0.71 | 0.78 | 2.18 | 2.48 | 0.30 | 36.0 | 24.1 | 23.4 | 16.4 | 699,686 |
| **North** | | | | | | | | | | | | |
| Haryana | 0.57 | 0.65 | 0.67 | 0.82 | 2.05 | 2.20 | 0.15 | 35.1 | 26.8 | 25.3 | 12.7 | 21,654 |
| Himachal Pradesh | 0.50 | 0.69 | 0.62 | 0.93 | 1.88 | 2.15 | 0.28 | 42.9 | 22.8 | 29.9 | 4.5 | 9,929 |
| Jammu & Kashmir | 0.42 | 0.69 | 0.67 | 0.78 | 2.01 | 1.65 | -0.36 | 46.0 | 20.0 | 21.2 | 12.9 | 23,800 |
| Punjab | 0.44 | 0.57 | 0.59 | 0.89 | 1.62 | 1.44 | -0.18 | 40.3 | 28.0 | 25.9 | 5.8 | 19,484 |
| Rajasthan | 0.63 | 0.65 | 0.68 | 0.89 | 2.40 | 2.68 | 0.28 | 33.3 | 31.1 | 27.2 | 8.4 | 41,965 |
| Uttarakhand | 0.51 | 0.70 | 0.67 | 0.82 | 2.07 | 2.13 | 0.06 | 41.2 | 21.7 | 24.7 | 12.4 | 17,300 |
| **Central** | | | | | | | | | | | | |
| Chhattisgarh | 0.55 | 0.71 | 0.66 | 0.82 | 2.23 | 2.31 | 0.08 | 38.3 | 22.2 | 26.5 | 13.1 | 25,172 |
| Madhya Pradesh | 0.60 | 0.73 | 0.68 | 0.78 | 2.32 | 2.55 | 0.23 | 34.7 | 21.8 | 26.8 | 16.7 | 62,803 |
| Uttar Pradesh | 0.59 | 0.72 | 0.69 | 0.82 | 2.74 | 2.64 | -0.10 | 37.0 | 22.7 | 26.0 | 14.3 | 97,661 |
| **East** | | | | | | | | | | | | |
| Bihar | 0.72 | 0.86 | 0.79 | 0.73 | 3.41 | 3.90 | 0.49 | 31.7 | 15.0 | 22.3 | 31.0 | 45,812 |
| Jharkhand | 0.65 | 0.78 | 0.73 | 0.73 | 2.55 | 2.90 | 0.36 | 33.1 | 19.1 | 23.7 | 24.1 | 29,046 |
| Odisha | 0.57 | 0.70 | 0.72 | 0.62 | 2.05 | 1.94 | -0.11 | 32.3 | 20.6 | 19.0 | 28.1 | 33,721 |
| West Bengal | 0.67 | 0.59 | 0.69 | 0.62 | 1.77 | 1.83 | 0.06 | 22.8 | 29.2 | 20.8 | 27.2 | 17,668 |
| **Northeast** | | | | | | | | | | | | |
| Arunachal Pradesh | 0.59 | 0.80 | 0.67 | 0.68 | 2.10 | 2.32 | 0.21 | 34.1 | 14.6 | 26.3 | 25.1 | 14,294 |
| Assam | 0.63 | 0.67 | 0.67 | 0.68 | 2.21 | 2.08 | -0.13 | 27.8 | 24.4 | 24.4 | 23.5 | 28,447 |
| Manipur | 0.52 | 0.85 | 0.69 | 0.75 | 2.61 | 2.51 | -0.10 | 44.6 | 10.9 | 25.3 | 19.2 | 13,593 |
| Meghalaya | 0.54 | 0.83 | 0.72 | 0.75 | 3.04 | 2.69 | -0.35 | 43.4 | 13.2 | 23.3 | 20.1 | 9,202 |
| Mizoram | 0.40 | 0.78 | 0.62 | 0.89 | 2.27 | 1.86 | -0.41 | 52.2 | 14.2 | 27.0 | 6.7 | 12,279 |
| Nagaland | 0.48 | 0.82 | 0.65 | 0.93 | 2.74 | 2.61 | -0.13 | 51.7 | 13.5 | 29.7 | 5.1 | 10,790 |
| Sikkim | 0.44 | 0.73 | 0.53 | 0.68 | 1.17 | 1.24 | 0.06 | 38.1 | 14.6 | 29.5 | 17.8 | 5,293 |
| Tripura | 0.63 | 0.64 | 0.64 | 0.62 | 1.68 | 1.74 | 0.06 | 24.9 | 24.1 | 24.6 | 26.4 | 4,804 |
| **West** | | | | | | | | | | | | |
| Goa | 0.45 | 0.82 | 0.72 | 0.73 | 1.66 | 2.09 | 0.43 | 48.9 | 11.8 | 20.1 | 19.3 | 1,696 |
| Gujarat | 0.57 | 0.74 | 0.66 | 0.89 | 2.03 | 2.71 | 0.68 | 40.0 | 21.8 | 29.7 | 8.5 | 22,932 |
| Maharashtra | 0.55 | 0.68 | 0.72 | 0.82 | 1.87 | 2.37 | 0.50 | 39.4 | 25.6 | 21.7 | 13.3 | 29,460 |
| **South** | | | | | | | | | | | | |
| Andhra Pradesh | 0.59 | 0.69 | 0.70 | 0.89 | 1.83 | 2.75 | 0.93 | 37.8 | 27.5 | 26.2 | 8.6 | 10,428 |

*(Continued)*

**Table 1.** (Continued)

| National/Region State | Indices | | | | Fertility | | | % contribution of determinant to fertility reduction from total fecundity§ | | | | Number of Women: Unweighted Ns |
|---|---|---|---|---|---|---|---|---|---|---|---|---|
| | Marriage (Cm) | Contraception (Cc) | Induced Abortion (Ca) | Postpartum Infecundability (Ci) | TFRa* | TFRe** | Residual† | Marriage | Contraception | Abortion | Postpartum Infecundability | |
| Karnataka | 0.55 | 0.75 | 0.71 | 0.82 | 1.80 | 2.62 | 0.82 | 42.1 | 19.9 | 23.8 | 14.2 | 26,291 |
| Kerala | 0.49 | 0.74 | 0.67 | 0.85 | 1.56 | 2.28 | 0.72 | 45.5 | 18.9 | 25.4 | 10.3 | 11,033 |
| Tamil Nadu | 0.50 | 0.73 | 0.69 | 0.93 | 1.70 | 2.59 | 0.89 | 48.2 | 21.5 | 25.3 | 5.0 | 28,820 |
| Telangana | 0.55 | 0.74 | 0.71 | 0.82 | 1.78 | 2.58 | 0.81 | 41.6 | 20.5 | 23.8 | 14.1 | 7,567 |

* Actual Total Fertility Rates.

** Estimated Total Fertility Rates, calculated as a function of the four indices and Total Fecundity. Total Fecundity or TF is based on an external source as recommended (the NFHS-3); it is 10.9 children per woman, and this value is applied nationally, for all states and population subgroups.

† Residual = The difference between TFRe and TFRa.

§ The proportionate reduction in fertility (from the Total Fecundity Rate to the actual Total Fertility Rate) that is attributable to each proximate determinant.

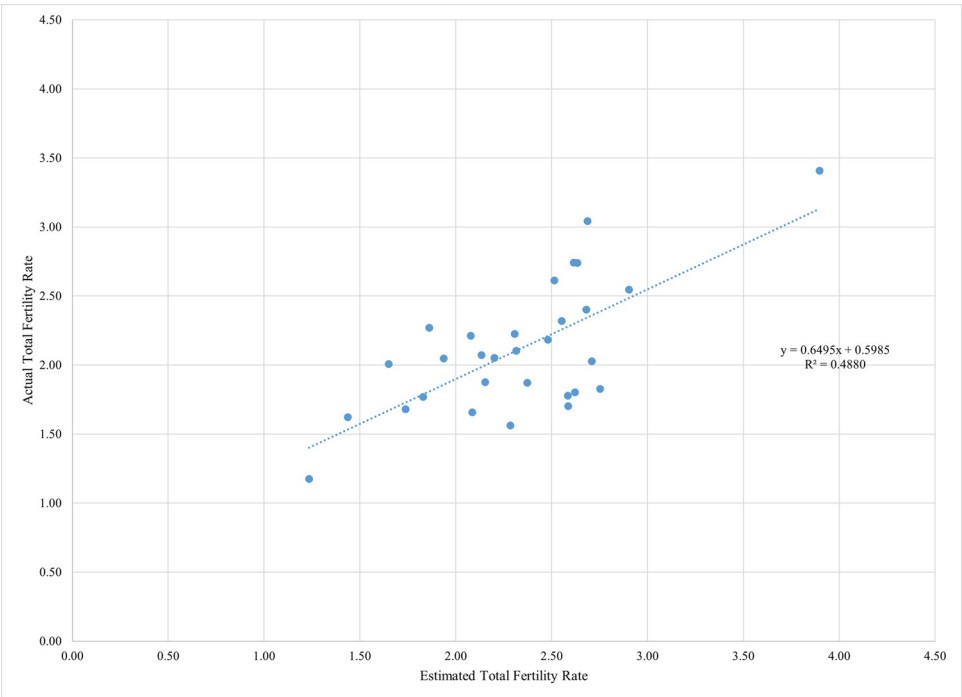

**Fig 1. A moderate positive correlation is observed between actual and estimated total fertility rates, India 2015–16.**

Contraception and induced abortion have similar indices (0.7 and 0.71) and each contributes almost one-fourth of the reduction from *TF*; postpartum infecundability has the largest index (0.78) and makes the smallest contribution (16%) to fertility reduction from *TF*.

Application of the model that includes the four key proximate determinants results in a difference of 0.30 births at the national level between the actual TFR (2.18) and the predicted TFR (2.48): this is a very good fit, given that the average residual or gap was 0.7 births in a study that applied the model to 36 countries [6]. The model produces residuals varying across the 29 states from low levels (-.13 to .08 births) in 10 states to relatively high levels (0.81 to 0.93 births) in four states, all in the South. The regression line between actual and predicted state level TFR has a moderate fit ($R^2$ = 49%) (Fig 1). Nationally, and in all states (Table 1), and for most of the socioeconomic subgroups (Tables 3–6), the estimated or predicted TFR is larger than the actual TFR, an expected pattern given that there are other unmeasured factors that may influence fertility levels, apart from the four measured factors [4].

## Impact of proximate determinants at the state level

Overall, the proportional contribution of marriage accounted for 36 percent of fertility reduction at the national level, and the contribution was similar (within +/- five percentage points) in 14 out of 29 states (Table 1). Three states where marriage accounted for substantially less than the national average proportion of fertility reduction are West Bengal (23 percent), Tripura (25 percent) and Assam (28 percent). In several states, marriage had a notably larger than average contribution (43%-52%)–Himachal Pradesh and Jammu & Kashmir in the North, four of the seven states in the Northeast (Manipur, Meghalaya, Mizoram and Nagaland), and two states in the South (Kerala and Tamil Nadu). As expected, the proportionate impact of contraception was at least 5% lower than the national average of 24 percent in most of the

states with TFRs substantially above the national average (2.6 or higher), including Bihar, Jharkhand, Meghalaya, Manipur and Nagaland. However, in four other states with low to average TFRs (1.7–2.3), contraception also had a significantly below average contribution (14–17 percent in Arunachal Pradesh, Mizoram, Goa, and Sikkim). Contraception had a higher-than-average contribution only in Rajasthan (31 percent). In almost all states, the contribution of abortion was similar to the national level (23 percent): Exceptions where abortion had a higher-than-average contribution are Gujarat, Himachal Pradesh, Nagaland and Sikkim (29–30 percent).

The national impact of postpartum infecundability was 16 percent. It was higher than the national average in all four states in the eastern region (Bihar, Jharkhand, Odisha and West Bengal (24–31 percent), and in Arunachal Pradesh, Assam and Tripura in the northeast region (24–26 percent). The contribution of postpartum infecundability was lower-than-average (4–9 percent in nine states), offset in most cases by marriage postponement either alone or in conjunction with abortion and/or contraception.

## Impact of proximate determinants across age groups: National level

Estimates of the proximate determinants show that the relative role of marriage at the national level declined with age, leveling out between age groups 30–34 and 45–49 (Table 2 and Fig 2). The contribution of marriage as a proximate determinant was highest among younger women as expected, accounting for 84 percent of fertility reduction for the age group 15–19 and 42 percent for women age 20–24. This is consistent with the substantial rise in the age at marriage among women since 2000 and low contraceptive use among younger women. The effect of contraception rose sharply with age up to age 25–34 before declining slowly at older ages (Table 2 and Fig 2).

Contraceptive use was the dominant determinant among women ages 25–49, accounting for 42 percent or more of fertility reduction. Induced abortion's impact on fertility increased with age, and it had the second largest effect on fertility reduction across all age groups: It was second to marriage (and somewhat larger than contraception) at ages 15–24, and second to contraception at ages 25–49 (Table 2 and Fig 2). At the national level, the impact of postpartum infecundability rose sharply from ages 15–19 (5 percent) to 20–24 (20 percent) after which it steadily declined (Table 2 and Fig 2). The highest contributions (20–21 percent) were observed in the main childbearing age range (20–29). In sum, while marriage exhibited the

**Table 2. Estimates of the percent contribution of four key proximate determinants to fertility reduction by age-group, actual ASFR and number of women: India 2015–16.**

| Age group | % contribution of determinant to fertility reduction from total fecundity§ | | | | Age-specific fertility rate | Number of Women: Unweighted Ns |
|---|---|---|---|---|---|---|
| | Marriage | Contraception | Abortion | Postpartum Infecundability | | |
| 15–19 | 83.5 | 2.9 | 8.3 | 5.2 | 51 | 124,878 |
| 20–24 | 42.2 | 17.3 | 20.4 | 20.1 | 184 | 122,955 |
| 25–29 | 9.9 | 42.2 | 26.9 | 21.1 | 128 | 115,076 |
| 30–34 | 3.5 | 52.8 | 25.9 | 17.8 | 51 | 97,048 |
| 35–39 | 3.1 | 52.8 | 30.1 | 14.1 | 17 | 90,433 |
| 40–44 | 2.9 | 47.5 | 37.8 | 11.7 | 4 | 76,627 |
| 45–49 | 3.1 | 49.7 | 36.8 | 10.5 | 1 | 72,669 |

§ The proportionate reduction in fertility (from the Total Fecundity Rate to the actual Total Fertility Rate) that is attributable to each proximate determinant.

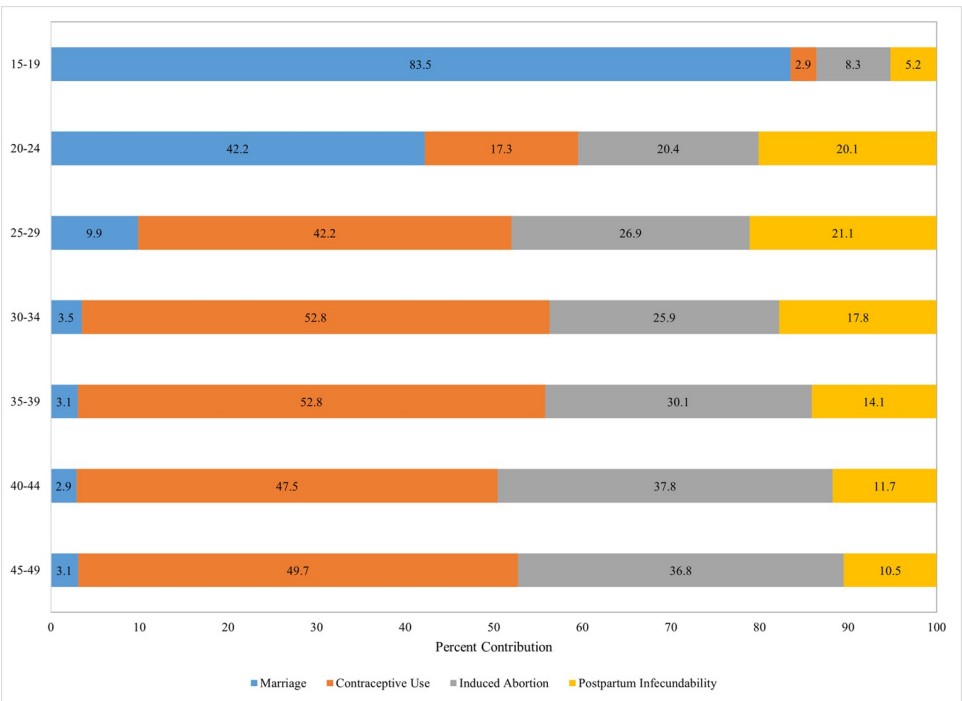

**Fig 2. Contraception and abortion are the leading fertility inhibiting factors among women above age 25, India, 2015–16.**

most dominant role in reducing fertility in adolescence and in early adulthood, contraceptive use assumed the most important role from the middle through the end of the reproductive age-range, with abortion making the next largest contribution, from ages 25–29 onwards (Fig 2).

## Impact of proximate determinants across age groups: State level

Patterns of contributions of the four key proximate determinants by age across states are for the most part similar to those found at the national level. As is true at the national level, the effect of marriage on fertility reduction was the highest in the 15–19 age group followed by the 20–24 age group in virtually all 29 states, and a sharp decline above ages 20–24, leveling off at ages 25–29 and higher at a very low level (S2 Table; in addition, S3 Table presents the indices themselves). A few exceptions to this pattern are found in the Northeastern states of Manipur, Nagaland and Mizoram, where the effect of marriage declined more gradually or stabilized above national levels (Meghalaya, 13–16 percent up to ages 45–49).

Similar to the national pattern, the impact of contraception in all states increased with age before plateauing around age 30–39 and declining only slightly among women in their 40s. Contraceptive use had the highest proportional contribution at ages 30–34 in 12 states, and at 35–39 in 14 states. At age 25–29, the effect of contraception was stronger than the impact of marriage in 24 states; the exceptions were four states in the Northeast and Goa in the West, all states with a relatively high age at first marriage among women.

The proportional contribution of induced abortion to fertility reduction generally increased with age with small variations at the state-level, mirroring the pattern observed at the national level. Similar to the national pattern, in most states the effect of postpartum infecundability on fertility reduction was lowest at ages 15–19 and increased up to ages 20–24, then declined

above age 25–29. The analysis of age-specific patterns in the impact of this factor across states is constrained by suppression of estimates of this index due to the low fertility rates at older ages, and the resulting small sample sizes of women who had recent births.

This study finds a strong association between actual and model predicted age-specific fertility rates. A regression line was fitted taking actual age-specific fertility rate as outcome variable and estimated age-specific fertility rate as explanatory variable for 178 observations (taking only those age groups for which all four key fertility indices and ASFRs could be estimated from 29 states and the country as a whole), as well as within each region's group of states. The results show a very close fit between predicted and actual ASFRs nationally ($R^2 = 0.91$) and for regions ($R^2$ ranging between 0.83 and 0.97).

## Impact of proximate determinants among sociodemographic subgroups: National level

The overall national pattern of relative contributions among all women ages 15–49—marriage having the largest contribution (36 percent), followed by abortion and contraception (about 24 percent each), and with postpartum infecundability having the lowest contribution (16 percent)—is found among most socioeconomic subgroups at the national level (Tables 3–6 and Fig 3; in addition, the indices for the four determinants by state and subgroup are presented in S4 Table). However, marriage had a greater impact (43 percent) and postpartum infecundability had a lower impact (9 percent) among urban women compared to rural women and compared to the national pattern. Similarly, among women with 5–9 years of schooling, the relative contribution of the four determinants were very similar to the national pattern. However, the other two education subgroups differed substantially from the national pattern: (a) among the least educated women, marriage and abortion had smaller impacts (25 and 18

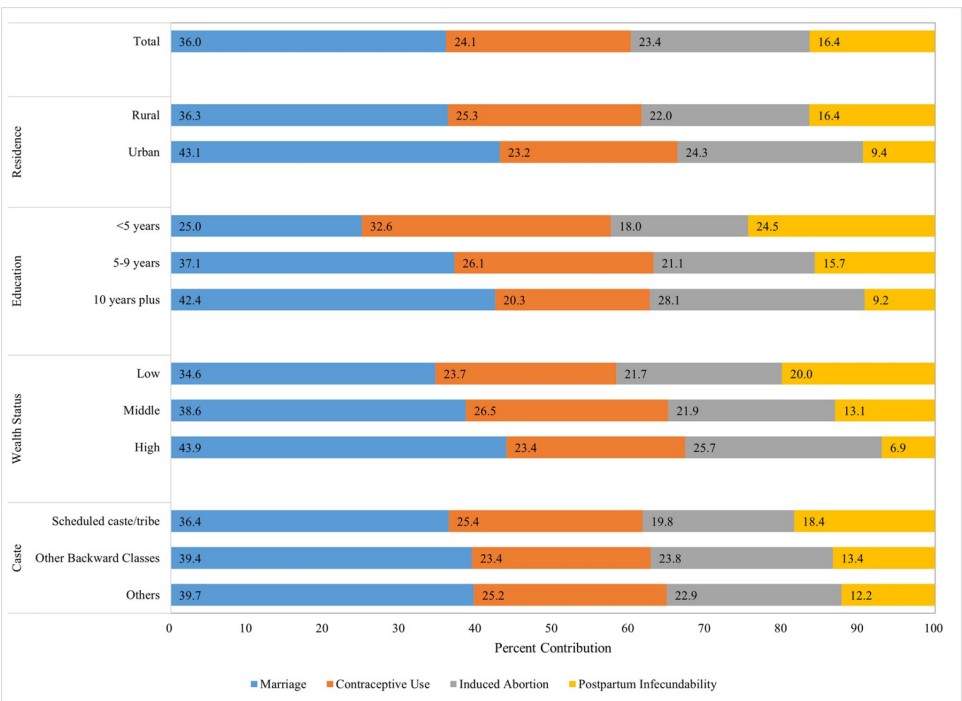

**Fig 3. Fertility inhibiting effect of marriage increases, while that of postpartum infecundability decreases with improvements in sociodemographic status, India, 2015–16.**

**Table 3. Estimates of the percent contribution of four key proximate determinants to fertility reduction, actual TFR, estimated TFR and residual by residence: National and by state, India 2015–16.**

| Region State | Residence | % contribution of determinant to fertility reduction from total fecundity§ | | | | Fertility | | |
|---|---|---|---|---|---|---|---|---|
| | | Marriage | Contraception | Abortion | Postpartum Infecundability | TFRa* | TFRe** | Residual† |
| **National** | | 36.0 | 24.1 | 23.4 | 16.4 | 2.18 | 2.48 | 0.30 |
| | Urban | 43.1 | 23.2 | 24.3 | 9.4 | 1.82 | 2.00 | 0.18 |
| | Rural | 36.3 | 25.3 | 22.0 | 16.4 | 2.45 | 2.50 | 0.05 |
| **North** | | | | | | | | |
| Haryana | | 35.1 | 26.8 | 25.3 | 12.7 | 2.05 | 2.20 | 0.15 |
| | Urban | 36.7 | 23.1 | 28.2 | {12.0} | 1.90 | {2.0} | {.10} |
| | Rural | 35.8 | 31.3 | 20.2 | 12.8 | 2.36 | 2.20 | -0.16 |
| Himachal Pradesh | | 42.9 | 22.8 | 29.9 | 4.5 | 1.88 | 2.15 | 0.28 |
| | Urban | a | a | a | b | 1.49 | b | b |
| | Rural | 45.7 | 23.6 | 26.3 | 4.4 | 1.97 | 2.10 | 0.13 |
| Jammu & Kashmir | | 46.0 | 20.0 | 21.2 | 12.9 | 2.01 | 1.65 | -0.36 |
| | Urban | 53.7 | 21.4 | 19.5 | 5.4 | 1.65 | 1.20 | -0.45 |
| | Rural | 45.0 | 18.7 | 21.6 | 14.8 | 2.26 | 1.60 | -0.66 |
| Punjab | | 40.3 | 28.0 | 25.9 | 5.8 | 1.62 | 1.44 | -0.18 |
| | Urban | 42.5 | 28.9 | 22.7 | 5.8 | 1.70 | 1.40 | -0.30 |
| | Rural | 40.8 | 27.9 | 25.7 | 5.6 | 1.73 | 1.30 | -0.43 |
| Rajasthan | | 33.3 | 31.1 | 27.2 | 8.4 | 2.40 | 2.68 | 0.28 |
| | Urban | 40.0 | 28.5 | 27.3 | 4.3 | 2.01 | 2.00 | -0.01 |
| | Rural | 34.3 | 31.8 | 25.6 | 8.4 | 2.57 | 2.70 | 0.13 |
| Uttarakhand | | 41.2 | 21.7 | 24.7 | 12.4 | 2.07 | 2.13 | 0.06 |
| | Urban | 46.7 | 21.4 | 25.1 | 6.8 | 1.87 | 1.90 | 0.03 |
| | Rural | 41.1 | 21.0 | 21.6 | 16.3 | 2.27 | 1.90 | -0.37 |
| **Central** | | | | | | | | |
| Chhattisgarh | | 38.3 | 22.2 | 26.5 | 13.1 | 2.23 | 2.31 | 0.08 |
| | Urban | 43.4 | 21.6 | 28.4 | 6.7 | 1.87 | 1.90 | 0.03 |
| | Rural | 40.9 | 21.8 | 24.4 | 12.9 | 2.45 | 2.30 | -0.15 |
| Madhya Pradesh | | 34.7 | 21.8 | 26.8 | 16.7 | 2.32 | 2.55 | 0.23 |
| | Urban | 42.2 | 19.3 | 29.0 | 9.5 | 2.01 | 2.00 | -0.01 |
| | Rural | 34.9 | 23.6 | 22.4 | 19.1 | 2.51 | 2.50 | -0.01 |
| | | Marriage | Contraception | Abortion | Postpartum Infecundability | TFRa* | TFRe** | Residual† |
| Uttar Pradesh | | 37.0 | 22.7 | 26.0 | 14.3 | 2.74 | 2.64 | -0.10 |
| | Urban | 46.1 | 23.7 | 23.5 | 6.7 | 2.13 | 1.90 | -0.23 |
| | Rural | 38.4 | 20.3 | 27.4 | 14.0 | 2.97 | 2.50 | -0.47 |
| **East** | | | | | | | | |
| Bihar | | 31.7 | 15.0 | 22.3 | 31.0 | 3.41 | 3.90 | 0.49 |
| | Urban | 45.2 | 15.4 | 22.1 | 17.3 | 2.46 | 2.70 | 0.24 |
| | Rural | 35.7 | 13.5 | 21.3 | 29.5 | 3.61 | 3.70 | 0.09 |
| Jharkhand | | 33.1 | 19.1 | 23.7 | 24.1 | 2.55 | 2.90 | 0.36 |
| | Urban | 45.2 | 14.7 | 22.1 | 17.9 | 1.79 | 1.80 | 0.01 |
| | Rural | 31.6 | 19.4 | 22.1 | 26.8 | 2.89 | 2.90 | 0.01 |
| Odisha | | 32.3 | 20.6 | 19.0 | 28.1 | 2.05 | 1.94 | -0.11 |
| | Urban | 36.2 | 21.4 | 19.3 | 23.1 | 1.76 | 1.50 | -0.26 |
| | Rural | 34.0 | 21.3 | 17.6 | 27.1 | 2.16 | 1.80 | -0.36 |
| West Bengal | | 22.8 | 29.2 | 20.8 | 27.2 | 1.77 | 1.83 | 0.06 |

*(Continued)*

**Table 3.** (Continued)

| | | Marriage | Contraception | Abortion | Postpartum Infecundability | TFRa* | TFRe** | Residual† |
|---|---|---|---|---|---|---|---|---|
| | Urban | 29.7 | 28.3 | 24.2 | {17.8} | 1.60 | {1.50} | {-.10} |
| | Rural | 22.2 | 32.4 | 17.3 | 28.0 | 1.87 | 1.70 | -0.17 |
| **Northeast** | | | | | | | | |
| Arunachal Pradesh | | 34.1 | 14.6 | 26.3 | 25.1 | 2.10 | 2.32 | 0.21 |
| | Urban | 39.0 | 10.7 | 26.8 | 23.5 | 1.82 | 1.80 | -0.02 |
| | Rural | 36.8 | 15.7 | 22.1 | 25.4 | 2.34 | 2.40 | 0.06 |
| Assam | | 27.8 | 24.4 | 24.4 | 23.5 | 2.21 | 2.08 | -0.13 |
| | Urban | 34.5 | 23.9 | 28.0 | {13.7} | 1.49 | {1.40} | {-.09} |
| | Rural | 29.0 | 24.8 | 23.3 | 22.9 | 2.40 | 2.00 | -0.40 |
| Manipur | | 44.6 | 10.9 | 25.3 | 19.2 | 2.61 | 2.51 | -0.10 |
| | Urban | 46.0 | 9.7 | 20.9 | 23.4 | 2.22 | 1.80 | -0.42 |
| | Rural | 47.1 | 11.4 | 24.1 | 17.4 | 3.06 | 2.70 | -0.36 |
| Meghalaya | | 43.4 | 13.2 | 23.3 | 20.1 | 3.04 | 2.69 | -0.35 |
| | Urban | 52.1 | 11.0 | 26.9 | {9.9} | 1.73 | {1.40} | {-.33} |
| | Rural | 44.8 | 12.1 | 19.9 | 23.2 | 3.63 | 2.80 | -0.83 |
| Mizoram | | 52.2 | 14.2 | 27.0 | 6.7 | 2.27 | 1.86 | -0.41 |
| | Urban | 55.6 | 15.2 | 23.6 | 5.6 | 2.08 | 1.30 | -0.78 |
| | Rural | 50.5 | 14.3 | 24.9 | 10.2 | 2.85 | 2.20 | -0.65 |
| | | **Marriage** | **Contraception** | **Abortion** | **Postpartum Infecundability** | **TFRa*** | **TFRe**** | **Residual†** |
| Nagaland | | 51.7 | 13.5 | 29.7 | 5.1 | 2.74 | 2.61 | -0.13 |
| | Urban | 55.3 | 12.6 | 30.8 | 1.3 | 1.86 | 1.60 | -0.26 |
| | Rural | 52.1 | 13.2 | 25.6 | 9.0 | 3.49 | 3.00 | -0.49 |
| Sikkim | | 38.1 | 14.6 | 29.5 | 17.8 | 1.17 | 1.24 | 0.06 |
| | Urban | 38.9 | 10.8 | 29.6 | b | 1.18 | b | b |
| | Rural | 42.0 | 17.4 | 26.5 | {14.1} | 1.26 | {1.10} | {-.16} |
| Tripura | | 24.9 | 24.1 | 24.6 | 26.4 | 1.68 | 1.74 | 0.06 |
| | Urban | 26.8 | 24.7 | 21.4 | b | 1.44 | b | b |
| | Rural | 26.9 | 26.4 | 22.5 | {24.3} | 1.85 | {1.70} | {-.15} |
| **West** | | | | | | | | |
| Goa | | 48.9 | 11.8 | 20.1 | 19.3 | 1.66 | 2.09 | 0.43 |
| | Urban | a | a | a | b | 1.77 | b | b |
| | Rural | a | a | a | b | 1.57 | b | b |
| Gujarat | | 40.0 | 21.8 | 29.7 | 8.5 | 2.03 | 2.71 | 0.68 |
| | Urban | 43.4 | 21.5 | 33.5 | 1.6 | 1.87 | 2.40 | 0.53 |
| | Rural | 41.6 | 22.3 | 24.6 | 11.4 | 2.19 | 2.70 | 0.51 |
| Maharashtra | | 39.4 | 25.6 | 21.7 | 13.3 | 1.87 | 2.37 | 0.50 |
| | Urban | 43.6 | 24.5 | 22.7 | 9.2 | 1.76 | 1.90 | 0.14 |
| | Rural | 37.7 | 29.9 | 18.4 | 14.0 | 2.11 | 2.60 | 0.49 |
| **South** | | | | | | | | |
| Andhra Pradesh | | 37.8 | 27.5 | 26.2 | 8.6 | 1.83 | 2.75 | 0.93 |
| | Urban | 40.1 | 26.8 | 28.8 | b | 1.60 | b | b |
| | Rural | 35.3 | 33.8 | 19.0 | b | 2.04 | b | b |
| Karnataka | | 42.1 | 19.9 | 23.8 | 14.2 | 1.80 | 2.62 | 0.82 |
| | Urban | 43.8 | 17.7 | 23.7 | 14.9 | 1.75 | 2.10 | 0.35 |
| | Rural | 41.6 | 24.9 | 18.9 | 14.6 | 1.99 | 2.70 | 0.71 |
| Kerala | | 45.5 | 18.9 | 25.4 | 10.3 | 1.56 | 2.28 | 0.72 |
| | Urban | 49.0 | 17.7 | 26.0 | {7.4} | 1.60 | {2.20} | {.60} |
| | Rural | 46.5 | 17.5 | 19.9 | 16.2 | 1.56 | 1.90 | 0.34 |
| Tamil Nadu | | 48.2 | 21.5 | 25.3 | 5.0 | 1.70 | 2.59 | 0.89 |

(*Continued*)

**Table 3.** (Continued)

|  |  |  |  |  |  |  |  |  |
|---|---|---|---|---|---|---|---|---|
|  | Urban | 50.7 | 21.8 | 25.9 | 1.6 | 1.62 | 2.20 | 0.58 |
|  | Rural | 47.3 | 24.3 | 20.4 | 8.0 | 1.95 | 2.50 | 0.55 |
|  |  | **Marriage** | **Contraception** | **Abortion** | **Postpartum Infecundability** | **TFRa**\* | **TFRe**\*\* | **Residual†** |
| Telangana |  | 41.6 | 20.5 | 23.8 | 14.1 | 1.78 | 2.58 | 0.81 |
|  | Urban | 44.9 | 21.4 | 26.3 | b | 1.81 | b | b |
|  | Rural | 37.9 | 26.9 | 18.0 | b | 1.97 | b | b |

§ The proportionate reduction in fertility (from the Total Fecundity Rate to the actual Total Fertility Rate) that is attributable to each proximate determinant.

\* Actual Total Fertility Rates.

\*\* Estimated Total Fertility Rates.

† Residual = The difference between TFRe and TFRa.

a = cell count less than 50 cases (unweighted).

[] = cell count between 50–100 cases (unweighted).

b = no. of women who have given birth in past 3 years less than 25 cases (unweighted).

{} = no. of women who have given birth in past 3 years between 25–50 cases (unweighted).

percent, respectively); and contraception and postpartum infecundability had greater impacts (33 and 25 percent, respectively); (b) among the highest educated group, this pattern was reversed. The lower two wealth groups were similar to the national pattern in relative importance of the four determinants, with small variations. However, among women with the highest wealth status, marriage was more important and postpartum infecundability was less important than average. The three caste/tribe subgroups were similar to the national pattern in terms of the relative contributions of the four proximate determinants, with small variations. Notably, only in the case of the lowest wealth status group was the predicted TFR found to be less than the actual TFR.

## Impact of proximate determinants across sociodemographic subgroups: State level

**Residence.** Similar to the national pattern, for most subgroups in states, marriage remained the greatest contributor to reducing fertility, while postpartum infecundability had the least impact; contraception and abortion typically had similar levels of impact, often alternating between the second and third position of importance (Table 3). However, clear exceptions are notable. For seven states (Bihar, Meghalaya, Manipur, Nagaland, Arunachal Pradesh, Mizoram, and Sikkim), contraception had a substantially weaker effect on fertility than abortion for both rural and urban residents; this pattern is found among urban but not rural women, in Chhattisgarh, Gujarat, Jharkhand and Kerala. In most of these cases, it is observed that the reduction in the effect of contraception was matched by an increase in the contribution of postpartum infecundability. Among urban women the effect of postpartum infecundability was as high as 23–24 percent in Arunachal Pradesh, Manipur, and Odisha, and among rural women it was in the similar range (25–30 percent in Arunachal Pradesh, Bihar, Jharkhand, Odisha, and West Bengal–compared to national averages of 9% and 16% for urban and rural women respectively.

**Education.** Again, similar to the national pattern, for most states, the contributions of marriage and abortion increased and those of contraception and postpartum infecundability declined, as the level of education rose, with a small number of variations from this pattern (Table 4). Among women with less than 5 years of schooling in the 20 states that met the

**Table 4. Estimates of the percent contribution of four key proximate determinants to fertility reduction, actual TFR, estimated TFR and residual by education: National and by state, India 2015–16.**

| Region State | Education | % contribution of determinant to fertility reduction from total fecundity§ | | | | Fertility | | |
|---|---|---|---|---|---|---|---|---|
| | | Marriage | Contraception | Abortion | Postpartum Infecundability | TFRa* | TFRe** | Residual† |
| **National** | | 36.0 | 24.1 | 23.4 | 16.4 | 2.18 | 2.48 | 0.30 |
| | <5 years | 25.0 | 32.6 | 18.0 | 24.5 | 3.13 | 3.40 | 0.27 |
| | 5–9 years | 37.1 | 26.1 | 21.1 | 15.7 | 2.30 | 2.30 | 0.00 |
| | 10 years plus | 42.4 | 20.3 | 28.1 | 9.2 | 1.84 | 1.90 | 0.06 |
| **North** | | | | | | | | |
| Haryana | | 35.1 | 26.8 | 25.3 | 12.7 | 2.05 | 2.20 | 0.15 |
| | <5 years | 21.4 | 34.3 | 22.6 | 21.7 | 3.49 | 3.60 | 0.11 |
| | 5–9 years | 36.7 | 29.1 | 19.3 | {14.9} | 2.34 | {2.10} | {-.24} |
| | 10 years plus | 37.5 | 25.0 | 28.5 | {9.0} | 1.77 | {1.80} | {.03} |
| Himachal Pradesh | | 42.9 | 22.8 | 29.9 | 4.5 | 1.88 | 2.15 | 0.28 |
| | <5 years | a | a | a | b | 2.95 | b | b |
| | 5–9 years | 45.4 | 27.7 | 22.3 | b | 2.30 | b | b |
| | 10 years plus | 40.9 | 19.3 | 35.9 | 3.9 | 2.00 | 1.70 | -0.30 |
| Jammu & Kashmir | | 46.0 | 20.0 | 21.2 | 12.9 | 2.01 | 1.65 | -0.36 |
| | <5 years | 34.3 | 27.9 | 17.1 | 20.6 | 2.98 | 2.80 | -0.18 |
| | 5–9 years | 46.6 | 20.4 | 20.5 | 12.5 | 2.18 | 1.60 | -0.58 |
| | 10 years plus | 51.5 | 18.3 | 22.6 | 7.6 | 1.87 | 1.30 | -0.57 |
| Punjab | | 40.3 | 28.0 | 25.9 | 5.8 | 1.62 | 1.44 | -0.18 |
| | <5 years | 26.1 | 44.7 | 15.3 | {13.9} | 2.78 | {2.50} | {-.28} |
| | 5–9 years | 39.6 | 30.5 | 23.7 | {6.2} | 2.10 | {1.60} | {-.5} |
| | 10 years plus | 41.7 | 25.3 | 29.8 | 3.2 | 1.54 | 1.20 | -0.34 |
| | | Marriage | Contraception | Abortion | Postpartum Infecundability | TFRa* | TFRe** | Residual† |
| Rajasthan | | 33.3 | 31.1 | 27.2 | 8.4 | 2.40 | 2.68 | 0.28 |
| | <5 years | 24.7 | 41.7 | 19.3 | 14.3 | 3.17 | 3.50 | 0.33 |
| | 5–9 years | 36.1 | 28.0 | 28.0 | 8.0 | 2.46 | 2.50 | 0.04 |
| | 10 years plus | 39.0 | 25.6 | 31.2 | 4.1 | 1.85 | 1.90 | 0.05 |
| Uttarakhand | | 41.2 | 21.7 | 24.7 | 12.4 | 2.07 | 2.13 | 0.06 |
| | <5 years | 30.9 | 28.1 | 20.8 | 20.3 | 3.20 | 3.30 | 0.10 |
| | 5–9 years | 43.6 | 21.7 | 21.9 | {12.8} | 2.36 | {2.2} | {-.16} |
| | 10 years plus | 44.6 | 19.0 | 25.9 | {10.5} | 1.77 | {1.60} | {-.17} |
| **Central** | | | | | | | | |
| Chhattisgarh | | 38.3 | 22.2 | 26.5 | 13.1 | 2.23 | 2.31 | 0.08 |
| | <5 years | 26.9 | 30.6 | 20.8 | 21.7 | 2.98 | 3.60 | 0.62 |
| | 5–9 years | 41.6 | 23.1 | 22.7 | 12.5 | 2.50 | 2.20 | -0.30 |
| | 10 years plus | 43.3 | 17.9 | 32.4 | {6.4} | 1.96 | {1.70} | {-.26} |
| Madhya Pradesh | | 34.7 | 21.8 | 26.8 | 16.7 | 2.32 | 2.55 | 0.23 |
| | <5 years | 20.9 | 36.2 | 16.5 | 26.3 | 3.17 | 3.70 | 0.53 |

(*Continued*)

**Table 4.** (Continued)

| | | Marriage | Contraception | Abortion | Postpartum Infecundability | TFRa* | TFRe** | Residual† |
|---|---|---|---|---|---|---|---|---|
| | 5–9 years | 37.1 | 21.4 | 25.8 | 15.7 | 2.40 | 2.30 | -0.10 |
| | 10 years plus | 41.6 | 16.4 | 33.2 | 8.8 | 1.91 | 1.70 | -0.21 |
| Uttar Pradesh | | 37.0 | 22.7 | 26.0 | 14.3 | 2.74 | 2.64 | -0.10 |
| | <5 years | 32.7 | 25.3 | 21.6 | 20.4 | 3.68 | 3.30 | -0.38 |
| | 5–9 years | 42.4 | 21.8 | 25.2 | 10.7 | 2.75 | 2.40 | -0.35 |
| | 10 years plus | 43.0 | 18.5 | 29.3 | 9.1 | 2.03 | 1.90 | -0.13 |
| **East** | | | | | | | | |
| Bihar | | 31.7 | 15.0 | 22.3 | 31.0 | 3.41 | 3.90 | 0.49 |
| | <5 years | 23.7 | 17.8 | 19.6 | 38.9 | 4.35 | 4.80 | 0.45 |
| | 5–9 years | 43.3 | 12.9 | 22.0 | 21.8 | 2.90 | 3.00 | 0.10 |
| | 10 years plus | 43.1 | 11.2 | 25.8 | 19.9 | 2.28 | 2.70 | 0.42 |
| | | **Marriage** | **Contraception** | **Abortion** | **Postpartum Infecundability** | **TFRa*** | **TFRe*** | **Residual†** |
| Jharkhand | | 33.1 | 19.1 | 23.7 | 24.1 | 2.55 | 2.90 | 0.36 |
| | <5 years | 22.6 | 25.2 | 18.6 | 33.6 | 3.32 | 3.80 | 0.48 |
| | 5–9 years | 36.8 | 18.6 | 20.4 | 24.2 | 2.57 | 2.50 | -0.07 |
| | 10 years plus | 40.4 | 13.9 | 28.6 | 17.1 | 1.98 | 2.10 | 0.12 |
| Odisha | | 32.3 | 20.6 | 19.0 | 28.1 | 2.05 | 1.94 | -0.11 |
| | <5 years | 24.7 | 27.7 | 14.0 | 33.7 | 2.72 | 2.60 | -0.12 |
| | 5–9 years | 32.5 | 21.7 | 19.4 | 26.4 | 2.19 | 1.90 | -0.29 |
| | 10 years plus | 38.9 | 17.8 | 21.7 | 21.6 | 1.67 | 1.30 | -0.37 |
| West Bengal | | 22.8 | 29.2 | 20.8 | 27.2 | 1.77 | 1.83 | 0.06 |
| | <5 years | 14.4 | 37.1 | 15.7 | 32.8 | 2.33 | 2.30 | -0.03 |
| | 5–9 years | 22.4 | 31.9 | 19.7 | {26.1} | 1.82 | {1.70} | {-.12} |
| | 10 years plus | 29.3 | 26.9 | 22.3 | {21.5} | 1.55 | {1.30} | {-.25} |
| **Northeast** | | | | | | | | |
| Arunachal Pradesh | | 34.1 | 14.6 | 26.3 | 25.1 | 2.10 | 2.32 | 0.21 |
| | <5 years | 26.2 | 19.7 | 20.3 | 33.8 | 3.12 | 3.50 | 0.38 |
| | 5–9 years | 37.1 | 16.2 | 23.3 | 23.4 | 2.16 | 2.10 | -0.06 |
| | 10 years plus | 39.5 | 9.5 | 30.4 | 20.7 | 1.73 | 1.70 | -0.03 |
| Assam | | 27.8 | 24.4 | 24.4 | 23.5 | 2.21 | 2.08 | -0.13 |
| | <5 years | 20.3 | 29.5 | 20.7 | 29.5 | 2.90 | 2.60 | -0.30 |
| | 5–9 years | 29.1 | 24.6 | 23.8 | 22.5 | 2.26 | 1.90 | -0.36 |
| | 10 years plus | 35.3 | 22.3 | 24.5 | 17.9 | 1.86 | 1.50 | -0.36 |
| Manipur | | 44.6 | 10.9 | 25.3 | 19.2 | 2.61 | 2.51 | -0.10 |
| | <5 years | 43.5 | 13.1 | 25.3 | 18.1 | 3.59 | 3.60 | 0.01 |
| | 5–9 years | 45.6 | 11.8 | 23.8 | 18.8 | 2.89 | 2.40 | -0.49 |
| | 10 years plus | 47.6 | 9.4 | 22.0 | 21.0 | 2.49 | 2.00 | -0.49 |
| | | **Marriage** | **Contraception** | **Abortion** | **Postpartum Infecundability** | **TFRa*** | **TFRe*** | **Residual†** |
| Meghalaya | | 43.4 | 13.2 | 23.3 | 20.1 | 3.04 | 2.69 | -0.35 |
| | <5 years | 37.4 | 16.0 | 20.6 | 26.1 | 4.67 | 3.70 | -0.97 |

(*Continued*)

**Table 4.** (Continued)

| | | Marriage | Contraception | Abortion | Postpartum Infecundability | TFRa* | TFRe** | Residual † |
|---|---|---|---|---|---|---|---|---|
| | 5–9 years | 45.7 | 13.5 | 21.4 | 19.4 | 3.24 | 2.60 | -0.64 |
| | 10 years plus | 49.5 | 8.6 | 20.5 | 21.4 | 2.08 | 1.50 | -0.58 |
| Mizoram | | 52.2 | 14.2 | 27.0 | 6.7 | 2.27 | 1.86 | -0.41 |
| | <5 years | 44.0 | 15.2 | 26.7 | 14.0 | 3.29 | 3.50 | 0.21 |
| | 5–9 years | 51.9 | 17.6 | 23.9 | 6.5 | 2.61 | 1.80 | -0.81 |
| | 10 years plus | 56.3 | 11.7 | 24.6 | 7.4 | 2.04 | 1.20 | -0.84 |
| Nagaland | | 51.7 | 13.5 | 29.7 | 5.1 | 2.74 | 2.61 | -0.13 |
| | <5 years | 38.3 | 19.3 | 26.0 | 16.3 | 4.43 | 4.10 | -0.33 |
| | 5–9 years | 53.7 | 13.8 | 27.5 | 5.1 | 2.82 | 2.60 | -0.22 |
| | 10 years plus | 58.6 | 10.3 | 27.5 | 3.6 | 2.09 | 1.40 | -0.69 |
| Sikkim | | 38.1 | 14.6 | 29.5 | 17.8 | 1.17 | 1.24 | 0.06 |
| | <5 years | [27.0] | [26.3] | [21.0] | b | 1.85 | [b] | [b] |
| | 5–9 years | 37.3 | 15.6 | 26.9 | b | 1.46 | b | b |
| | 10 years plus | 44.6 | 10.7 | 33.4 | b | 1.07 | b | b |
| Tripura | | 24.9 | 24.1 | 24.6 | 26.4 | 1.68 | 1.74 | 0.06 |
| | <5 years | 14.5 | 32.8 | 20.6 | b | 2.34 | b | b |
| | 5–9 years | 27.0 | 26.7 | 23.8 | b | 1.72 | b | b |
| | 10 years plus | 29.0 | 23.7 | 20.3 | b | 1.61 | b | b |
| **West** | | | | | | | | |
| Goa | | 48.9 | 11.8 | 20.1 | 19.3 | 1.66 | 2.09 | 0.43 |
| | <5 years | a | a | a | b | 3.08 | b | b |
| | 5–9 years | a | a | a | b | 1.90 | b | b |
| | 10 years plus | a | a | a | b | 1.68 | b | b |
| | | **Marriage** | **Contraception** | **Abortion** | **Postpartum Infecundability** | **TFRa*** | **TFRe**** | **Residual †** |
| Gujarat | | 40.0 | 21.8 | 29.7 | 8.5 | 2.03 | 2.71 | 0.68 |
| | <5 years | 27.6 | 31.2 | 18.4 | 22.9 | 2.82 | 3.80 | 0.98 |
| | 5–9 years | 39.3 | 23.3 | 28.9 | 8.5 | 2.29 | 2.70 | 0.41 |
| | 10 years plus | 44.4 | 17.7 | 37.9 | {0} | 1.58 | {1.90} | {.32} |
| Maharashtra | | 39.4 | 25.6 | 21.7 | 13.3 | 1.87 | 2.37 | 0.50 |
| | <5 years | 24.2 | 39.6 | 10.3 | 25.9 | 2.51 | 3.20 | 0.69 |
| | 5–9 years | 35.0 | 30.4 | 18.0 | {16.5} | 2.33 | {2.50} | {.17} |
| | 10 years plus | 42.5 | 22.3 | 28.6 | 6.6 | 1.78 | 1.80 | 0.02 |
| **South** | | | | | | | | |
| Andhra Pradesh | | 37.8 | 27.5 | 26.2 | 8.6 | 1.83 | 2.75 | 0.93 |
| | <5 years | 14.4 | 55.1 | 11.2 | b | 2.26 | b | b |
| | 5–9 years | 28.5 | 41.0 | 20.9 | b | 2.18 | b | b |
| | 10 years plus | 39.2 | 21.3 | 32.7 | b | 1.90 | b | b |
| Karnataka | | 42.1 | 19.9 | 23.8 | 14.2 | 1.80 | 2.62 | 0.82 |
| | <5 years | 25.3 | 35.6 | 10.2 | 28.9 | 2.27 | 3.60 | 1.33 |
| | 5–9 years | 37.8 | 26.2 | 21.3 | {14.7} | 2.18 | {2.70} | {.52} |

(*Continued*)

**Table 4.** (Continued)

| | | Marriage | Contraception | Abortion | Postpartum Infecundability | TFRa* | TFRe** | Residual† |
|---|---|---|---|---|---|---|---|---|
| | 10 years plus | 43.7 | 16.1 | 30.2 | 10.1 | 1.90 | 2.20 | 0.30 |
| Kerala | | 45.5 | 18.9 | 25.4 | 10.3 | 1.56 | 2.28 | 0.72 |
| | <5 years | a | a | a | b | 1.53 | b | b |
| | 5–9 years | [53.6] | [16.6] | [15.1] | {14.7} | 1.76 | {1.60} | {-.16} |
| | 10 years plus | 43.0 | 15.2 | 32.7 | 9.1 | 1.66 | 1.90 | 0.24 |
| Tamil Nadu | | 48.2 | 21.5 | 25.3 | 5.0 | 1.70 | 2.59 | 0.89 |
| | <5 years | 29.2 | 40.0 | 9.4 | {21.4} | 2.11 | {3.50} | {1.39} |
| | 5–9 years | 41.1 | 29.3 | 20.4 | {9.3} | 2.38 | {3.10} | {.72} |
| | 10 years plus | 46.1 | 18.9 | 30.7 | 4.3 | 1.82 | 2.00 | 0.18 |
| | | **Marriage** | **Contraception** | **Abortion** | **Postpartum Infecundability** | **TFRa*** | **TFRe**** | **Residual†** |
| Telangana | | 41.6 | 20.5 | 23.8 | 14.1 | 1.78 | 2.58 | 0.81 |
| | <5 years | 21.6 | 40.9 | 12.0 | b | 2.21 | b | b |
| | 5–9 years | 37.1 | 27.7 | 21.0 | b | 2.00 | b | b |
| | 10 years plus | 42.1 | 20.1 | 30.4 | b | 2.00 | b | b |

§ The proportionate reduction in fertility (from the Total Fecundity Rate to the actual Total Fertility Rate) that is attributable to each proximate determinant.

* Actual Total Fertility Rates.

** Estimated Total Fertility Rates.

† Residual = The difference between TFRe and TFRa.

a = cell count less than 50 cases (unweighted).

[] = cell count between 50–100 cases (unweighted).

b = no. of women who have given birth in past 3 years less than 25 cases (unweighted).

{} = no. of women who have given birth in past 3 years between 25–50 cases (unweighted).

minimum sample size criteria for estimating all four indices, similar to the national pattern, abortion often had the smallest impact; contraception was found to be the most prominent determinant in nine of these states, and marriage was the most prominent in seven of the 20 states. The pattern of effects of the proximate determinants was quite similar among women with 5–9 years of schooling: In all 15 states that met the minimum sample size for calculating all four indices for the 5–9 years of schooling subgroup, marriage had the strongest effect, as it did at the national level.

Nine of the 14 states that met sample size criteria for calculating all four indices for all three education groups saw an increase in the impact of abortion as education increased, and in the other five states of this group (all in the Northeast), abortion had a similar impact across the three education subgroups. In an additional 11 states for which the abortion index was estimated for all three education groups (but for which one other index could not be calculated given small sample size), all but one state (Tripura) also had a pattern of increasing impact of abortion as women's educational level increased.

**Wealth status.** Similar to the national pattern, as wealth status rose there was an increase in the contribution of marriage and a decrease in the contribution of postpartum infecundability to reducing fertility in 7 of the 10 states that met the sample size criteria for calculating these two indices for all three wealth subgroups (Table 5). In Arunachal Pradesh, one of the three states with adequate data that do not fit this national pattern, the contributions of

**Table 5. Estimates of the percent contribution of four key proximate determinants to fertility reduction, actual TFR, estimated TFR and residual by wealth status: National and by state, India 2015–16.**

| Region State | Wealth Status | % contribution of determinant to fertility reduction from total fecundity§ | | | | Fertility | | |
|---|---|---|---|---|---|---|---|---|
| | | Marriage | Contraception | Abortion | Postpartum Infecundability | TFRa* | TFRe** | Residual† |
| **National** | | 36.0 | 24.1 | 23.4 | 16.4 | 2.18 | 2.48 | 0.30 |
| | Low | 34.6 | 23.7 | 21.7 | 20.0 | 2.87 | 2.70 | -0.17 |
| | Middle | 38.6 | 26.5 | 21.9 | 13.1 | 2.10 | 2.30 | 0.20 |
| | High | 43.9 | 23.4 | 25.7 | 6.9 | 1.72 | 2.00 | 0.28 |
| **North** | | | | | | | | |
| Haryana | | 35.1 | 26.8 | 25.3 | 12.7 | 2.05 | 2.20 | 0.15 |
| | Low | 37.8 | 22.8 | 25.2 | 14.3 | 3.72 | 2.60 | -1.12 |
| | Middle | 37.5 | 28.5 | 18.1 | 15.9 | 2.51 | 2.40 | -0.11 |
| | High | 38.9 | 31.1 | 19.6 | 10.4 | 1.90 | 2.30 | 0.40 |
| Himachal Pradesh | | 42.9 | 22.8 | 29.9 | 4.5 | 1.88 | 2.15 | 0.28 |
| | Low | [45.4] | [33.1] | [21.5] | b | 2.49 | [b] | [b] |
| | Middle | 45.6 | 23.7 | 26.7 | {4.1} | 1.89 | {1.90} | {.01} |
| | High | 43.3 | 20.0 | 32.6 | 4.1 | 1.86 | 1.90 | 0.04 |
| Jammu & Kashmir | | 46.0 | 20.0 | 21.2 | 12.9 | 2.01 | 1.65 | -0.36 |
| | Low | 42.1 | 21.0 | 20.0 | 16.9 | 2.93 | 2.10 | -0.83 |
| | Middle | 49.5 | 19.6 | 18.6 | 12.2 | 1.95 | 1.50 | -0.45 |
| | High | 50.9 | 19.1 | 20.2 | 9.9 | 1.78 | 1.40 | -0.38 |
| Punjab | | 40.3 | 28.0 | 25.9 | 5.8 | 1.62 | 1.44 | -0.18 |
| | Low | [36.9] | [35.4] | [21.5] | b | 2.37 | [b] | [b] |
| | Middle | 40.9 | 29.5 | 19.3 | {10.3} | 2.05 | {1.50} | {-.55} |
| | High | 42.2 | 28.0 | 26.4 | 3.5 | 1.61 | 1.40 | -0.21 |
| Rajasthan | | 33.3 | 31.1 | 27.2 | 8.4 | 2.40 | 2.68 | 0.28 |
| | Low | 33.0 | 32.1 | 22.8 | 12.1 | 3.13 | 2.90 | -0.23 |
| | Middle | 35.6 | 29.9 | 26.6 | 7.9 | 2.31 | 2.40 | 0.09 |
| | High | 38.5 | 28.5 | 28.6 | 4.5 | 1.89 | 2.10 | 0.21 |
| Uttarakhand | | 41.2 | 21.7 | 24.7 | 12.4 | 2.07 | 2.13 | 0.06 |
| | Low | 43.4 | 22.9 | 18.9 | 14.8 | 2.81 | 2.10 | -0.71 |
| | Middle | 41.4 | 20.0 | 22.0 | 16.7 | 2.23 | 2.00 | -0.23 |
| | High | 46.0 | 21.8 | 25.5 | {6.8} | 1.80 | {1.90} | {.1} |
| **Central** | | | | | | | | |
| | | Marriage | Contraception | Abortion | Postpartum Infecundability | TFRa* | TFRe** | Residual† |
| Chhattisgarh | | 38.3 | 22.2 | 26.5 | 13.1 | 2.23 | 2.31 | 0.08 |
| | Low | 40.2 | 20.9 | 23.3 | 15.5 | 2.54 | 2.30 | -0.24 |
| | Middle | 42.1 | 23.2 | 24.9 | {9.9} | 2.18 | {2.10} | {-.08} |
| | High | 43.6 | 20.3 | 29.3 | 6.8 | 1.92 | 1.90 | -0.02 |
| Madhya Pradesh | | 34.7 | 21.8 | 26.8 | 16.7 | 2.32 | 2.55 | 0.23 |
| | Low | 34.5 | 23.8 | 21.7 | 20.0 | 2.78 | 2.70 | -0.08 |
| | Middle | 38.0 | 22.6 | 26.6 | 12.9 | 2.15 | 2.20 | 0.05 |
| | High | 40.8 | 19.2 | 30.7 | 9.3 | 1.83 | 1.90 | 0.07 |
| Uttar Pradesh | | 37.0 | 22.7 | 26.0 | 14.3 | 2.74 | 2.64 | -0.10 |
| | Low | 38.5 | 18.5 | 25.7 | 17.3 | 3.27 | 2.70 | -0.57 |
| | Middle | 40.9 | 22.5 | 26.1 | 10.6 | 2.60 | 2.40 | -0.20 |
| | High | 45.6 | 23.9 | 26.1 | 4.3 | 2.02 | 2.10 | 0.08 |

*(Continued)*

**Table 5.** (Continued)

| East | | | | | | | | |
|---|---|---|---|---|---|---|---|---|
| Bihar | | 31.7 | 15.0 | 22.3 | 31.0 | 3.41 | 3.90 | 0.49 |
| | Low | 34.3 | 13.6 | 20.9 | 31.2 | 3.95 | 3.90 | -0.05 |
| | Middle | 42.1 | 13.5 | 22.9 | 21.5 | 2.50 | 2.90 | 0.40 |
| | High | 48.4 | 12.4 | 25.7 | {13.5} | 2.03 | {2.40} | {.37} |
| Jharkhand | | 33.1 | 19.1 | 23.7 | 24.1 | 2.55 | 2.90 | 0.36 |
| | Low | 32.5 | 18.0 | 22.4 | 27.1 | 2.99 | 3.00 | 0.01 |
| | Middle | 37.4 | 21.5 | 22.6 | {18.5} | 2.14 | {2.40} | {.26} |
| | High | 45.5 | 14.2 | 26.5 | {13.8} | 1.66 | {1.90} | {.24} |
| Odisha | | 32.3 | 20.6 | 19.0 | 28.1 | 2.05 | 1.94 | -0.11 |
| | Low | 35.5 | 21.2 | 17.3 | 25.9 | 2.22 | 1.90 | -0.32 |
| | Middle | 32.7 | 21.2 | 17.8 | 28.4 | 2.02 | 1.80 | -0.22 |
| | High | 36.8 | 20.9 | 21.1 | {21.2} | 1.68 | {1.50} | {-.18} |
| West Bengal | | 22.8 | 29.2 | 20.8 | 27.2 | 1.77 | 1.83 | 0.06 |
| | Low | 22.5 | 31.4 | 19.1 | 26.9 | 2.02 | 1.80 | -0.22 |
| | Middle | 24.4 | 31.5 | 19.9 | {24.2} | 1.66 | {1.7} | {.04} |
| | High | 29.4 | 26.8 | 20.6 | {23.2} | 1.38 | {1.2} | {-.18} |
| **Northeast** | | | | | | | | |
| Arunachal Pradesh | | 34.1 | 14.6 | 26.3 | 25.1 | 2.10 | 2.32 | 0.21 |
| | Low | 35.6 | 17.9 | 21.1 | 25.5 | 2.87 | 2.70 | -0.17 |
| | Middle | 36.5 | 14.5 | 25.5 | 23.4 | 1.91 | 2.10 | 0.19 |
| | High | 40.1 | 9.2 | 26.9 | 23.8 | 1.70 | 1.60 | -0.10 |
| | | **Marriage** | **Contraception** | **Abortion** | **Postpartum Infecundability** | **TFRa*** | **TFRe**** | **Residual†** |
| Assam | | 27.8 | 24.4 | 24.4 | 23.5 | 2.21 | 2.08 | -0.13 |
| | Low | 27.5 | 25.2 | 23.1 | 24.2 | 2.67 | 2.20 | -0.47 |
| | Middle | 30.6 | 23.8 | 23.4 | 22.3 | 1.90 | 1.60 | -0.30 |
| | High | 35.5 | 22.4 | 25.2 | 16.8 | 1.48 | 1.30 | -0.18 |
| Manipur | | 44.6 | 10.9 | 25.3 | 19.2 | 2.61 | 2.51 | -0.10 |
| | Low | 50.6 | 9.8 | 25.0 | 14.6 | 3.12 | 2.70 | -0.42 |
| | Middle | 44.9 | 11.9 | 23.2 | 20.0 | 2.56 | 2.20 | -0.36 |
| | High | 45.6 | 9.3 | 20.0 | 25.2 | 2.51 | 2.00 | -0.51 |
| Meghalaya | | 43.4 | 13.2 | 23.3 | 20.1 | 3.04 | 2.69 | -0.35 |
| | Low | 40.0 | 13.3 | 22.6 | 24.1 | 4.44 | 3.40 | -1.04 |
| | Middle | 48.8 | 11.7 | 22.0 | 17.5 | 2.80 | 2.20 | -0.60 |
| | High | [50.8] | [10.8] | [19.5] | {18.9} | 1.58 | {1.20} | {-.38} |
| Mizoram | | 52.2 | 14.2 | 27.0 | 6.7 | 2.27 | 1.86 | -0.41 |
| | Low | 42.8 | 12.8 | 27.5 | 16.8 | 3.64 | 3.30 | -0.34 |
| | Middle | 53.2 | 17.9 | 24.5 | 4.3 | 2.71 | 2.10 | -0.61 |
| | High | 56.3 | 14.0 | 22.3 | 7.4 | 1.97 | 1.20 | -0.77 |
| Nagaland | | 51.7 | 13.5 | 29.7 | 5.1 | 2.74 | 2.61 | -0.13 |
| | Low | 46.4 | 15.1 | 27.5 | 11.0 | 4.59 | 3.70 | -0.89 |
| | Middle | 54.6 | 13.4 | 27.6 | 4.4 | 2.25 | 2.10 | -0.15 |
| | High | [56.6] | [11.6] | [30.6] | {1.2} | 1.67 | {1.40} | {-.27} |
| Sikkim | | 38.1 | 14.6 | 29.5 | 17.8 | 1.17 | 1.24 | 0.06 |
| | Low | a | a | a | b | 2.30 | b | b |
| | Middle | 41.3 | 16.9 | 27.7 | {14.2} | 1.29 | {1.20} | {-.09} |
| | High | [39.2] | [11.1] | [33.2] | b | 1.01 | [b] | [b] |

(*Continued*)

**Table 5.** (Continued)

| | | | | | | | | |
|---|---|---|---|---|---|---|---|---|
| Tripura | | 24.9 | 24.1 | 24.6 | 26.4 | 1.68 | 1.74 | 0.06 |
| | Low | 29.5 | 24.6 | 20.2 | {25.7} | 1.89 | {1.60} | {-.29} |
| | Middle | 23.2 | 28.8 | 17.8 | b | 1.63 | b | b |
| | High | [30.2] | [23.6] | [34.9] | b | 1.41 | [b] | [b] |
| **West** | | | | | | | | |
| Goa | | 48.9 | 11.8 | 20.1 | 19.3 | 1.66 | 2.09 | 0.43 |
| | Low | a | a | a | b | 2.77 | b | b |
| | Middle | a | a | a | b | 1.35 | b | b |
| | High | a | a | a | b | 1.72 | b | b |
| | | **Marriage** | **Contraception** | **Abortion** | **Postpartum Infecundability** | **TFRa*** | **TFRe**** | **Residual†** |
| Gujarat | | 40.0 | 21.8 | 29.7 | 8.5 | 2.03 | 2.71 | 0.68 |
| | Low | 37.9 | 21.4 | 22.2 | 18.5 | 2.73 | 2.90 | 0.17 |
| | Middle | 41.8 | 21.7 | 25.2 | 11.3 | 2.16 | 2.60 | 0.44 |
| | High | 43.0 | 21.3 | 35.7 | 0.0 | 1.66 | 2.30 | 0.64 |
| Maharashtra | | 39.4 | 25.6 | 21.7 | 13.3 | 1.87 | 2.37 | 0.50 |
| | Low | 37.5 | 29.1 | 15.2 | 18.3 | 2.36 | 2.30 | -0.06 |
| | Middle | 38.1 | 25.8 | 20.2 | {15.9} | 2.00 | {2.40} | {.4} |
| | High | 43.5 | 26.0 | 23.7 | 6.9 | 1.68 | 2.00 | 0.32 |
| **South** | | | | | | | | |
| Andhra Pradesh | | 37.8 | 27.5 | 26.2 | 8.6 | 1.83 | 2.75 | 0.93 |
| | Low | 32.1 | 33.5 | 18.8 | b | 2.30 | b | b |
| | Middle | 34.3 | 35.1 | 19.1 | b | 1.98 | b | b |
| | High | 41.1 | 23.6 | 31.0 | b | 1.61 | b | b |
| Karnataka | | 42.1 | 19.9 | 23.8 | 14.2 | 1.80 | 2.62 | 0.82 |
| | Low | 42.3 | 26.3 | 16.8 | {14.6} | 2.01 | {2.70} | {.69} |
| | Middle | 41.3 | 23.3 | 21.5 | 13.9 | 1.96 | 2.50 | 0.54 |
| | High | 44.1 | 16.3 | 26.9 | {12.7} | 1.73 | {2.20} | {.47} |
| Kerala | | 45.5 | 18.9 | 25.4 | 10.3 | 1.56 | 2.28 | 0.72 |
| | Low | a | a | a | b | 0.95 | b | b |
| | Middle | 46.0 | 18.7 | 19.5 | b | 1.68 | b | b |
| | High | 46.8 | 15.9 | 27.6 | 9.6 | 1.57 | 2.00 | 0.43 |
| Tamil Nadu | | 48.2 | 21.5 | 25.3 | 5.0 | 1.70 | 2.59 | 0.89 |
| | Low | 49.1 | 22.8 | 17.5 | {10.6} | 1.95 | {2.40} | {.45} |
| | Middle | 46.5 | 24.7 | 21.1 | {7.7} | 1.91 | {2.40} | {.49} |
| | High | 50.0 | 20.9 | 27.6 | 1.6 | 1.60 | 2.30 | 0.70 |
| Telangana | | 41.6 | 20.5 | 23.8 | 14.1 | 1.78 | 2.58 | 0.81 |
| | Low | 38.0 | 24.9 | 18.8 | b | 2.20 | b | b |
| | Middle | 38.9 | 26.1 | 18.7 | b | 1.90 | b | b |
| | High | 45.6 | 20.5 | 29.4 | b | 1.74 | b | b |

§ The proportionate reduction in fertility (from the Total Fecundity Rate to the actual Total Fertility Rate) that is attributable to each proximate determinant.

* Actual Total Fertility Rates.

** Estimated Total Fertility Rates.

† Residual = The difference between TFRe and TFRa.

a = cell count less than 50 cases (unweighted).

[] = cell count between 50–100 cases (unweighted).

b = no. of women who have given birth in past 3 years less than 25 cases (unweighted).

{} = no. of women who have given birth in past 3 years between 25–50 cases (unweighted).

marriage and postpartum infecundability to reducing fertility varied only slightly with increasing wealth status; in Haryana, the contribution of marriage did not vary with wealth status, though the contribution of postpartum infecundability decreased while moving from the middle to the highest wealth group. In Manipur, the national pattern was reversed: the contribution of marriage decreased from the lowest wealth tertile to the wealthier two groups, and the contribution of postpartum infecundability increased as wealth status increased. In 12 of the 21 that met the sample size criteria for calculating the contraception index for all three wealth subgroups, the role of contraception varied little across wealth status. In Arunachal Pradesh and Karnataka, the impact of contraception declined steadily from the lowest to the highest wealth status groups and in four other states (Andhra Pradesh, Jharkhand, West Bengal and Telangana), the impact of contraception declined, though not as systematically; it increased as wealth status rose in Haryana and Uttar Pradesh, and in Mizoram but not systematically. The contribution of abortion increased substantially (by 5% or more) as wealth status rose in 12 of the 21 states that met the sample size criteria for calculating the index of abortion for all three wealth subgroups. In Manipur and Mizoram, the contribution of abortion declined as wealth increased.

**Caste.** As caste advantage increased, the impact of marriage increased in 10 of 20 states with adequate sample size for estimating the impact of marriage for all three caste groups (Table 6). No clear pattern was found for the contribution of contraception to reducing fertility across caste subgroups in these same 20 states that had adequate data for this index and for abortion as well. The impact of abortion increased as caste advantage increased in seven of these 20 states: this increase was largest in Kerala and Gujarat (14 percentage points—from 12 percent for SCs/STs to 26 percent for Other castes in Kerala, and in Gujarat from 22 percent to 36 percent).

**Table 6. Estimates of the percent contribution of four key proximate determinants to fertility reduction, actual TFR, estimated TFR and residual by caste: National and by state, India 2015–16.**

| Region State | Caste | % contribution of determinant to fertility reduction from total fecundity§ | | | | Fertility | | |
|---|---|---|---|---|---|---|---|---|
| | | Marriage | Contraception | Abortion | Postpartum Infecundability | TFRa* | TFRe** | Residual† |
| **National** | | 36.0 | 24.1 | 23.4 | 16.4 | 2.18 | 2.48 | 0.30 |
| | Scheduled caste/tribe | 36.4 | 25.4 | 19.8 | 18.4 | 2.39 | 2.40 | 0.01 |
| | Other Backward Classes | 39.4 | 23.4 | 23.8 | 13.4 | 2.27 | 2.40 | 0.13 |
| | Others | 39.7 | 25.2 | 22.9 | 12.2 | 1.98 | 2.10 | 0.12 |
| **North** | | | | | | | | |
| Haryana | | 35.1 | 26.8 | 25.3 | 12.7 | 2.05 | 2.20 | 0.15 |
| | Scheduled caste/tribe | 35.9 | 27.7 | 21.4 | {15.0} | 2.35 | {2.20} | {-.15} |
| | Other Backward Classes | 34.4 | 29.5 | 23.5 | 12.6 | 2.23 | 2.20 | -0.03 |
| | Others | 39.1 | 24.9 | 26.6 | {9.4} | 1.88 | {2.00} | {.12} |
| Himachal Pradesh | | 42.9 | 22.8 | 29.9 | 4.5 | 1.88 | 2.15 | 0.28 |
| | Scheduled caste/tribe | 43.3 | 26.5 | 25.7 | {4.5} | 2.09 | {2.2} | {.11} |
| | Other Backward Classes | [45.5] | [18.3] | [25.6] | b | 1.72 | [b] | [b] |

*(Continued)*

**Table 6.** (Continued)

| | | Marriage | Contraception | Abortion | Postpartum Infecundability | TFRa* | TFRe** | Residual † |
|---|---|---|---|---|---|---|---|---|
| | Others | 46.8 | 21.8 | 29.9 | 1.4 | 1.87 | 2.00 | 0.13 |
| Jammu & Kashmir | | 46.0 | 20.0 | 21.2 | 12.9 | 2.01 | 1.65 | -0.36 |
| | Scheduled caste/tribe | 41.4 | 20.1 | 23.6 | 14.9 | 2.62 | 2.10 | -0.52 |
| | Other Backward Classes | [50.1] | [20.9] | [22.2] | b | 2.08 | [b] | [b] |
| | Others | 50.3 | 19.7 | 17.9 | 12.0 | 1.95 | 1.40 | -0.55 |
| Punjab | | 40.3 | 28.0 | 25.9 | 5.8 | 1.62 | 1.44 | -0.18 |
| | Scheduled caste/tribe | 41.5 | 29.0 | 23.7 | 5.9 | 1.93 | 1.50 | -0.43 |
| | Other Backward Classes | 41.1 | 29.5 | 23.3 | {6.1} | 1.71 | {1.60} | {-.11} |
| | Others | 41.3 | 27.3 | 26.1 | 5.3 | 1.52 | 1.20 | -0.32 |
| | | Marriage | Contraception | Abortion | Postpartum Infecundability | TFRa* | TFRe** | Residual † |
| Rajasthan | | 33.3 | 31.1 | 27.2 | 8.4 | 2.40 | 2.68 | 0.28 |
| | Scheduled caste/tribe | 34.0 | 30.4 | 24.1 | 11.4 | 2.70 | 2.70 | 0.00 |
| | Other Backward Classes | 34.2 | 31.8 | 25.9 | 8.0 | 2.41 | 2.50 | 0.09 |
| | Others | 40.4 | 27.1 | 25.6 | 7.0 | 2.01 | 2.00 | -0.01 |
| Uttarakhand | | 41.2 | 21.7 | 24.7 | 12.4 | 2.07 | 2.13 | 0.06 |
| | Scheduled caste/tribe | 44.4 | 19.2 | 20.4 | {15.9} | 2.05 | {1.90} | {-.15} |
| | Other Backward Classes | 44.8 | 24.8 | 25.7 | 4.8 | 2.50 | 2.40 | -0.10 |
| | Others | 42.8 | 19.4 | 22.4 | 15.5 | 1.97 | 1.80 | -0.17 |
| **Central** | | | | | | | | |
| Chhattisgarh | | 38.3 | 22.2 | 26.5 | 13.1 | 2.23 | 2.31 | 0.08 |
| | Scheduled caste/tribe | 39.7 | 21.4 | 25.7 | 13.2 | 2.52 | 2.30 | -0.22 |
| | Other Backward Classes | 43.1 | 22.3 | 22.4 | 12.2 | 2.17 | 2.10 | -0.07 |
| | Others | 44.5 | 19.4 | 27.3 | {8.8} | 1.90 | {1.70} | {-.20} |
| Madhya Pradesh | | 34.7 | 21.8 | 26.8 | 16.7 | 2.32 | 2.55 | 0.23 |
| | Scheduled caste/tribe | 36.6 | 23.9 | 22.8 | 16.7 | 2.63 | 2.50 | -0.13 |
| | Other Backward Classes | 36.9 | 22.2 | 25.0 | 15.9 | 2.28 | 2.40 | 0.12 |
| | Others | 40.2 | 19.2 | 28.8 | 11.9 | 1.98 | 2.00 | 0.02 |
| Uttar Pradesh | | 37.0 | 22.7 | 26.0 | 14.3 | 2.74 | 2.64 | -0.10 |
| | Scheduled caste/tribe | 38.4 | 20.8 | 23.7 | 17.1 | 3.12 | 2.60 | -0.52 |
| | Other Backward Classes | 41.1 | 21.8 | 26.2 | 10.9 | 2.76 | 2.50 | -0.26 |
| | Others | 44.7 | 22.1 | 26.0 | 7.2 | 2.31 | 2.10 | -0.21 |

(*Continued*)

**Table 6.** (Continued)

| East | | | | | | | | |
|---|---|---|---|---|---|---|---|---|
| Bihar | | 31.7 | 15.0 | 22.3 | 31.0 | 3.41 | 3.90 | 0.49 |
| | Scheduled caste/tribe | 33.6 | 14.5 | 20.0 | 31.9 | 4.00 | 4.00 | 0.00 |
| | Other Backward Classes | 36.5 | 14.6 | 20.1 | 28.9 | 3.41 | 3.60 | 0.19 |
| | Others | 44.2 | 11.6 | 22.2 | 22.0 | 2.89 | 3.00 | 0.11 |
| | | **Marriage** | **Contraception** | **Abortion** | **Postpartum Infecundability** | **TFRa***  | **TFRe**** | **Residual †** |
| Jharkhand | | 33.1 | 19.1 | 23.7 | 24.1 | 2.55 | 2.90 | 0.36 |
| | Scheduled caste/tribe | 34.8 | 16.1 | 23.2 | 25.9 | 2.69 | 2.80 | 0.11 |
| | Other Backward Classes | 35.1 | 21.2 | 20.9 | 22.8 | 2.59 | 2.70 | 0.11 |
| | Others | 44.8 | 16.2 | 23.5 | {15.5} | 2.16 | {2.30} | {.14} |
| Odisha | | 32.3 | 20.6 | 19.0 | 28.1 | 2.05 | 1.94 | -0.11 |
| | Scheduled caste/tribe | 33.4 | 23.0 | 16.7 | 26.8 | 2.37 | 2.00 | -0.37 |
| | Other Backward Classes | 33.9 | 20.7 | 16.8 | 28.6 | 1.90 | 1.60 | -0.30 |
| | Others | 38.7 | 17.8 | 20.3 | 23.2 | 1.84 | 1.50 | -0.34 |
| West Bengal | | 22.8 | 29.2 | 20.8 | 27.2 | 1.77 | 1.83 | 0.06 |
| | Scheduled caste/tribe | 22.9 | 31.6 | 15.2 | {30.3} | 1.76 | {1.60} | {-.16} |
| | Other Backward Classes | 27.6 | 31.2 | 20.9 | b | 1.69 | b | b |
| | Others | 24.3 | 29.5 | 20.7 | 25.5 | 1.81 | 1.60 | -0.21 |
| **Northeast** | | | | | | | | |
| Arunachal Pradesh | | 34.1 | 14.6 | 26.3 | 25.1 | 2.10 | 2.32 | 0.21 |
| | Scheduled caste/tribe | 36.4 | 11.7 | 26.7 | 25.2 | 2.13 | 2.00 | -0.13 |
| | Other Backward Classes | [35.2] | [20.0] | [23.2] | b | 2.22 | [b] | [b] |
| | Others | 35.9 | 22.6 | 23.3 | b | 2.40 | b | b |
| Assam | | 27.8 | 24.4 | 24.4 | 23.5 | 2.21 | 2.08 | -0.13 |
| | Scheduled caste/tribe | 31.3 | 23.5 | 22.1 | 23.1 | 2.07 | 1.80 | -0.27 |
| | Other Backward Classes | 33.0 | 22.6 | 23.7 | 20.7 | 1.90 | 1.70 | -0.20 |
| | Others | 28.0 | 25.9 | 24.1 | 22.1 | 2.57 | 2.20 | -0.37 |
| Manipur | | 44.6 | 10.9 | 25.3 | 19.2 | 2.61 | 2.51 | -0.10 |
| | Scheduled caste/tribe | 51.1 | 10.5 | 23.5 | 14.9 | 3.25 | 2.80 | -0.45 |
| | Other Backward Classes | 47.4 | 7.5 | 19.3 | 25.8 | 2.17 | 1.70 | -0.47 |
| | Others | 41.5 | 11.3 | 21.4 | 25.8 | 2.54 | 2.10 | -0.44 |

(*Continued*)

**Table 6.** (Continued)

| | | Marriage | Contraception | Abortion | Postpartum Infecundability | TFRa* | TFRe** | Residual† |
|---|---|---|---|---|---|---|---|---|
| Meghalaya | | 43.4 | 13.2 | 23.3 | 20.1 | 3.04 | 2.69 | -0.35 |
| | Scheduled caste/tribe | 47.7 | 10.6 | 22.8 | 18.8 | 3.24 | 2.40 | -0.84 |
| | Other Backward Classes | a | a | a | b | 1.08 | b | b |
| | Others | [34.6] | [24.2] | [19.7] | b | 2.82 | [b] | [b] |
| Mizoram | | 52.2 | 14.2 | 27.0 | 6.7 | 2.27 | 1.86 | -0.41 |
| | Scheduled caste/tribe | 55.4 | 14.9 | 23.2 | 6.5 | 2.42 | 1.80 | -0.62 |
| | Other Backward Classes | a | a | a | b | 0.98 | b | b |
| | Others | a | a | a | b | 2.58 | b | b |
| Nagaland | | 51.7 | 13.5 | 29.7 | 5.1 | 2.74 | 2.61 | -0.13 |
| | Scheduled caste/tribe | 55.3 | 12.3 | 27.7 | 4.7 | 2.85 | 2.30 | -0.55 |
| | Other Backward Classes | a | a | a | b | 0.86 | b | b |
| | Others | a | a | a | b | 2.58 | b | b |
| Sikkim | | 38.1 | 14.6 | 29.5 | 17.8 | 1.17 | 1.24 | 0.06 |
| | Scheduled caste/tribe | 41.2 | 15.6 | 28.3 | {15.0} | 1.18 | {1.00} | {-.18} |
| | Other Backward Classes | 39.6 | 17.6 | 28.6 | b | 1.19 | b | b |
| | Others | [41.5] | [11.4] | [28.3] | b | 1.36 | [b] | [b] |
| Tripura | | 24.9 | 24.1 | 24.6 | 26.4 | 1.68 | 1.74 | 0.06 |
| | Scheduled caste/tribe | 28.4 | 26.2 | 20.3 | {25.1} | 1.77 | {1.60} | {-.17} |
| | Other Backward Classes | 26.6 | 24.4 | 18.9 | b | 1.75 | b | b |
| | Others | 24.7 | 26.1 | 26.4 | b | 1.67 | b | b |
| **West** | | | | | | | | |
| Goa | | 48.9 | 11.8 | 20.1 | 19.3 | 1.66 | 2.09 | 0.43 |
| | Scheduled caste/tribe | a | a | a | b | 1.38 | b | b |
| | Other Backward Classes | a | a | a | b | 1.66 | b | b |
| | Others | a | a | a | b | 1.80 | b | b |
| | | Marriage | Contraception | Abortion | Postpartum Infecundability | TFRa* | TFRe** | Residual† |
| Gujarat | | 40.0 | 21.8 | 29.7 | 8.5 | 2.03 | 2.71 | 0.68 |
| | Scheduled caste/tribe | 40.1 | 23.2 | 22.9 | 13.8 | 2.28 | 2.50 | 0.22 |
| | Other Backward Classes | 40.4 | 22.1 | 29.3 | 8.3 | 2.07 | 2.60 | 0.53 |
| | Others | 46.5 | 19.9 | 33.6 | 0.0 | 1.71 | 2.20 | 0.49 |

(*Continued*)

**Table 6.** (Continued)

| | | Marriage | Contraception | Abortion | Postpartum Infecundability | TFRa* | TFRe** | Residual† |
|---|---|---|---|---|---|---|---|---|
| Maharashtra | | 39.4 | 25.6 | 21.7 | 13.3 | 1.87 | 2.37 | 0.50 |
| | Scheduled caste/tribe | 37.7 | 27.0 | 19.6 | 15.7 | 2.09 | 2.30 | 0.21 |
| | Other Backward Classes | 43.9 | 26.1 | 20.2 | {9.8} | 1.78 | {2.10} | {.32} |
| | Others | 41.1 | 27.5 | 21.4 | 10.1 | 1.91 | 2.20 | 0.29 |
| **South** | | | | | | | | |
| Andhra Pradesh | | 37.8 | 27.5 | 26.2 | 8.6 | 1.83 | 2.75 | 0.93 |
| | Scheduled caste/tribe | 34.7 | 28.9 | 19.8 | b | 2.05 | b | b |
| | Other Backward Classes | 35.4 | 33.9 | 22.6 | b | 1.90 | b | b |
| | Others | 43.1 | 27.2 | 29.7 | b | 1.71 | b | b |
| Karnataka | | 42.1 | 19.9 | 23.8 | 14.2 | 1.80 | 2.62 | 0.82 |
| | Scheduled caste/tribe | 42.2 | 23.1 | 20.6 | {14.1} | 1.80 | {2.60} | {.80} |
| | Other Backward Classes | 43.8 | 22.2 | 23.2 | 10.9 | 1.93 | 2.50 | 0.57 |
| | Others | 42.4 | 18.7 | 20.8 | b | 1.92 | b | b |
| Kerala | | 45.5 | 18.9 | 25.4 | 10.3 | 1.56 | 2.28 | 0.72 |
| | Scheduled caste/tribe | 40.1 | 17.4 | 11.9 | b | 1.63 | b | b |
| | Other Backward Classes | 45.6 | 18.9 | 23.4 | 12.1 | 1.60 | 2.00 | 0.40 |
| | Others | 51.5 | 15.3 | 26.2 | {7.0} | 1.52 | {2.00} | {.48} |
| Tamil Nadu | | 48.2 | 21.5 | 25.3 | 5.0 | 1.70 | 2.59 | 0.89 |
| | Scheduled caste/tribe | 47.2 | 22.9 | 19.5 | {10.4} | 1.84 | {2.30} | {.46} |
| | Other Backward Classes | 48.4 | 22.4 | 24.6 | 4.6 | 1.76 | 2.30 | 0.54 |
| | Others | a | a | a | b | 1.87 | b | b |
| | | **Marriage** | **Contraception** | **Abortion** | **Postpartum Infecundability** | **TFRa*** | **TFRe**** | **Residual†** |
| Telangana | | 41.6 | 20.5 | 23.8 | 14.1 | 1.78 | 2.58 | 0.81 |
| | Scheduled caste/tribe | 41.9 | 23.7 | 20.4 | b | 1.82 | b | b |
| | Other Backward Classes | 39.9 | 25.1 | 21.4 | b | 1.88 | b | b |
| | Others | 47.7 | 22.9 | 24.8 | b | 2.05 | b | b |

§ The proportionate reduction in fertility (from the Total Fecundity Rate to the actual Total Fertility Rate) that is attributable to each proximate determinant.

* Actual Total Fertility Rates.

** Estimated Total Fertility Rates.

† Residual = The difference between TFRe and TFRa.

a = cell count less than 50 cases (unweighted).

[] = cell count between 50–100 cases (unweighted).

b = no. of women who have given birth in past 3 years less than 25 cases (unweighted).

{} = no. of women who have given birth in past 3 years between 25–50 cases (unweighted).

Among six out of eight states that met the sample size criteria for calculating the index of postpartum infecundability for all three caste subgroups, the only notable pattern was a substantial decline in the contribution of this index from the largest impact among the most disadvantaged subgroup (SCs/STs) to the smallest impact for the least disadvantaged group ("Other" castes). In the case of Manipur, the SCs/STs subgroup showed a much lesser fertility impact of postpartum infecundability than the other two subgroups; and in Assam, this impact was similar across caste groups.

## Discussion

This article analyses differences in the contribution of four key determinants—marriage, contraceptive use, induced abortion and postpartum infecundability—to average family size nationally, across states and among population subgroups in India. Building on earlier research [9], the current study focuses on the NFHS-4 (conducted in 2015–16) and incorporates abortion as a determinant at all levels of the analysis, using estimates of abortion incidence available for 2015, consistent with the timing of data from the NFHS-4 [14]. The earlier study was only able to incorporate abortion at the national level, given data available at that time. Additionally, the current study provides estimates of the four determinants for population subgroups according to key characteristics of women—place of residence, age, educational attainment, household wealth status, and caste—providing a means of assessing differences and potential inequities among population subgroups. The current study covers all 29 states, possible because the NFHS-4 had a much larger sample size than prior rounds of the NFHS.

The model is highly predictive of the actual level of fertility: At the national level, the difference is only 0.3 birth, compared to larger differences generally found (0.7 births and more) [6]. In addition, the regression coefficient between predicted and actual age-specific fertility rates is very high nationally and within each region ($R^2$ of 0.8 and higher). The results show that the four key proximate determinants predict the actual Total Fertility Rate very well, nationally and for most states and population subgroups. The remaining gaps between the predicted and the actual TFRs nationally are most likely due to unmeasured factors that influence fertility and underestimation of the impact of one or more of the four determinants included in the model.

Where larger differences are found between predicted and actual TFRs, unmeasured factors that explain them can be identified. For example, the likely reason for the higher-than-average residuals in the South is the higher prevalence of primary and secondary infertility, which is well documented [26–29]. When the five southern states (Andhra Pradesh, Karnataka, Kerala, Tamil Nadu, and Karnataka) are removed, $R^2$ increases from 49% to 72%. Another example is the case of two states (Meghalaya and Mizoram), where marriage has a larger than average impact (43% and 52% respectively compared to the national average of 36%, S2 Table): The factor contributing to this result is marriage customs. In both states, the prevalence of single mothers, forced marriage, polygamy, and separate living arrangements of husbands and wives in certain tribes mean that the index of marriage will not fully capture the effects of marriage on fertility [30–32]. A third example is the impact of migration resulting in separation of spouses, therefore reducing the risk of pregnancy within marriage [33]. Our results show a pattern of larger residuals for less educated compared to more educated women: the prevalence of male migration as unskilled and semi-skilled laborers among lower income, less educated families [34] may contribute to this pattern since it is likely to be associated with greater prevalence of prolonged abstinence among less educated women. The prevalent pattern of a decline in the contribution of contraception as education rises is unexpected, given that the opposite pattern

(increase in contraceptive use with rising education) is common in other countries [35]. However, factors that may contribute to a different pattern in India are need for effective reversible methods to space births—more common among younger women and therefore among the better educated—and the relatively narrow availability of such methods [36]; there is also some evidence of a preference among more educated women for traditional contraceptive methods and their ability to effectively use these methods [37, 38].

These results suggest that overall, India is moving towards homogeneity in the relative contribution of the four proximate determinants of fertility, as differences in the level of fertility across states and across subgroups within states narrow. Apart from educational subgroups, there are no systematic differences among socio-demographic subgroups at the national or state level, from the national pattern of the contribution of the four determinants. In addition, the contribution of the four factors were similar to the national pattern, across groupings of states by region and level of current fertility (S5 Table).

The results also show that, despite some variations, all four proximate factors are important determinants of the current level of fertility. The steady increase in the average age at first marriage [39–41] reflects broad social changes, including increasing proportions living in cities, rising educational attainment, social policies that encourage families to postpone marriage of their girl children until at least age 18, and changing values that support women being in the workforce and having an increased role in decision-making within the family. These changes are likely to be accompanied by a rise in women's status and increased ability to implement their fertility preferences. These same broad societal trends lead to an increased preference for smaller families and for controlling the spacing of births, which in turn is likely to result in an increased demand for contraception and abortion, strengthening the contribution of these two proximate determinants among states and population groups that are more urban, more highly educated and economically better off. The results show that women in all states and across all age and socioeconomic subgroups are relying on both contraception and abortion as means of fertility control, even though the relative importance of these two factors differs somewhat across groups and states. The reasons for similarities and differences across subgroups and states are likely to be wide-ranging and include accessibility and affordability of these two SRH services, as well as women's knowledge, perceptions, and preferences. The former group of reasons is likely to be more important for poor, less educated, rural, and lower caste women (groups for which contraception plays a smaller than average role), and the latter type of reasons appear to be relevant for some of the more advantaged population subgroups (for example abortion often has a larger contribution than contraception for these groups).

The finding that the contribution of abortion is often close to (and occasionally larger) than that of contraception in explaining current fertility levels supports other research documenting barriers that women face in obtaining good quality contraceptive services [36, 42]. A high proportion of contraceptive users rely on private sector providers–one third of sterilization users and 56% of users of other modern methods [43]. Given that private sector providers charge for services, and are concentrated in urban areas, this may also be a barrier for low income and rural women. It is also possible that, lacking acceptable effective short-term contraceptive methods, some women may prefer to use abortion as their means of spacing births or controlling family size—and the results suggest that this is likely the case for younger women. However, the fact that among women of higher wealth status, contraception has a smaller role than abortion in several states and an equal role in many other states suggests that in addition, poor quality of contraceptive services is also likely to be an important factor.

Having a choice of methods with good quality counselling and information is far from universal in India. In fact, less than one-third of clients reported receiving information about key factors affecting method choice and only half of these clients were told about the side effects of

the method they chose [44]. Poor quality of services is likely connected with high levels of discontinuation of method use: About 10–20% of women who started using the pill, IUD and injectables in the five years before interview, that is, about 25%-50% of all who discontinued use [2] stopped within 12 months because of method related reasons such as concerns about health and side-effects, wanting a more effective method and other method-related reasons (including lack of access, distance, cost and inconvenience of using the method). The clinical quality of contraceptive care is also lacking: A study in Uttar Pradesh (public and private facilities) and Bihar (public facilities) found that only 62% of facilities that offered sterilization services followed all required components of infection prevention and provider adherence to infection prevention practices occurred in only 68% of female sterilization procedures [45].

Our results show that abortion is an important means of fertility control in most states and subgroups. Abortion is legal under broad criteria in India, but access to safe services remains poor [14, 46, 47] and information provided to clients on MA use is generally inadequate [48, 49]. These findings reinforce the need for improvements in access to comprehensive abortion care.

The study has some data limitations that must be kept in mind. The analysis is for married women, which is a limitation in terms of population level conclusions, though the findings are relevant for the married women. It was not feasible to include unmarried women because survey data on sexual activity among unmarried women is likely to be highly underreported. Some studies have shown relatively low levels of sexual activity among unmarried young women [50]. While this suggests that the contribution of sexually active unmarried adolescents and young women to need for contraception and abortion may be relatively small in terms of magnitude, more studies and improved study designs are needed to better document the sexual and reproductive health and behaviours of this extremely vulnerable and at-risk group. Sample size limitations for some states (and for all Union Territories) and for many demographic and socio-economic subgroups meant that we were not able to present estimates for them. Preliminary analyses of NFHS-5 recently published for 22 states and union territories suggest that contraceptive use may be underestimated in a few states in the NFHS-4 survey. Six states show exceptionally steep increases between NFHS-4 (2015–16) and NFHS-5 (2019–20), suggesting that contraceptive use may be underreported in NFHS-4 (39); in addition, it is also possible that contraceptive use is over-reported in NFHS-5, or that both factors contributed to the observed difference.

## Policy and program recommendations

Based on this study's findings and their implications, we recommend a few critical strategies to improve access to contraceptive and safe abortion services in India. These will help couples in exercising and achieving their sexual reproductive health and rights as laid out India's Sustainable Development Goals [51, 52].

• Improve public education on both contraception and abortion.

a. Expand the use of existing outreach and public education programs under the National Health Mission (NHM) to provide information on both contraceptive and safe abortion services. It is efficient, effective, and feasible to implement existing media campaigns about contraceptive and abortion more widely and build on them by developing additional materials. One such mass media campaign was initiated by the NHM in 2014 to disseminate information regarding the legality and availability of induced abortion at public and registered private sites. This and similar campaigns should be continued.

b. Broaden the sources providing education by including community level providers: Accredited Social Health Activists (ASHAs) and Auxiliary Nurse Midwifes (ANMs) are often women's first point of contact with the health system. They provide contraceptive counseling and some contraceptive services. In addition, although these providers do not perform abortions, they can expand the range of their services to facilitate the early determination of pregnancy, provide counseling on decision making regarding the pregnancy and provide guidance on abortion services including where to obtain safe abortion.

- Improve the availability, accessibility, and quality of contraceptive services.

a. Increase provision of accurate information on contraceptive methods for spacing and stopping, including relative effectiveness and common side effects of each method. This can be done through popular communication channels for all people of reproductive age and by targeting specific groups (such family planning clinic clients and postpartum and post abortion patients). It is also critical that comprehensive sexuality education is provided to both in school and out of school adolescents.

b. Ensure that users are provided with a choice of methods, both short-term and longer-term reversible methods as well as sterilization.

c. Improve the quality of contraceptive services. Recent data from the NFHS-5 (2019–21) survey shows that districts that shown improvements in quality of services gained most on modern method use. Giving appropriate and adequate information, providing follow-up care and treating clients well will go a long way to ensure adoption and continuation of all methods, specifically, newly introduced spacing methods (e.g., injectables and post-partum IUD).

d. Monitor and ensure that coercive practices are eliminated with particular attention to sterilization. Contraceptive coercion is still practiced. For example, some providers insist that women who fit certain characteristics (e.g., married with two children) must accept tubal ligation as a condition for providing her with an abortion. Also, evidence suggests that women were being sterilized without being told that the procedure means an end to their childbearing and many women still receive financial incentives for sterilization [53].

- Improve the availability, accessibility, and quality of abortion care.

a. Expand the capacity of public health facilities, especially at the primary health center level to provide medical methods of abortion, vacuum aspiration and postabortion care by increasing the number of trained staff and the availability of equipment and drugs. This would improve access to vulnerable groups including rural, poor, young and unmarried women, and other disadvantaged groups.

b. Strengthen existing campaigns to widely disseminate information regarding the legality and availability of induced abortion services at public and registered private sites. Information and instructions on appropriate use of medical abortion drugs (including correct regimen, what to expect and where to go in case of complications) by means of clearly worded text in appropriate languages and use of pictorials. Recognizing that most abortions take place outside facilities, helplines should be set up to assist women seeking medical abortion and helpline numbers should be prominently displayed on combi-packs and at pharmacies.

c. Ensure that abortion care is provided confidentially, that providers treat patients with respect and are non-judgmental in their attitude to clients seeking abortion care, as emphasized in the MTP Amendment Act, 2021.

d. Require that the curricula for training abortion providers at all levels include updated information about the country's abortion laws (especially important in view of the passage of the MTP Amendment in 2021), and about women's rights to obtain abortion and sexual and reproductive healthcare, more broadly.

e. Provide mechanisms for regular monitoring to ensure that health care providers and other facility staff do not impose unnecessary limitations on abortion provision. This is necessary to ensure implementation of the provision under the MTP Amendment of 2021 that allows unmarried women to obtain abortion.

- Improve policies regarding early marriage and breastfeeding.

a. Continue and improve on initiatives such as *Dhanalakshmi Yojana* (a conditional cash transfer program was that launched by the central government in 2008 to encourage parents to delay marriage of girls to 18 or older and to educate girls*)* and *Sukanya Samridhi Yojana*, a second program that was launched in 2015 with a similar goal. Such programs are important given their potential to reduce early marriage and thereby improve women's decision-making regarding timing and number of children they will have.

b. Continue to implement an existing, important element in government maternal health services for postpartum women, that is, to support exclusive breastfeeding for six months after delivery.

- Conduct new studies to fill evidence gaps.

a. New and more comprehensive estimates of the incidence of abortion are needed, including estimates for each state. Independent and reliable estimates of abortion incidence, available at regular intervals, are needed to improve state governments' ability to address gaps in and barriers to abortion-related services.

b. Both public and private service providers need to be sensitized about the importance of keeping records on services provided. Improved and comprehensively implemented data collection systems and training provided on how to collect these data correctly and confidentially is likely to improve ability to effectively assess and meet service needs.

c. Better quality data on unmarried women's sexual and reproductive behaviors are also needed to effectively include the needs of young, unmarried women in effort to improve women's sexual and reproductive health and rights.

## Conclusion

Acknowledging that being free to decide how many children to have is a human right, it follows that access to sexual and reproductive health information and services is essential. Given that both contraception and abortion are important means in all states and across all population groups for managing fertility and avoiding unwanted pregnancy and births, supportive and non-coercive policies, adequate resources and mechanisms to monitor quality and access to these essential sexual and reproductive health services, must be improved. This study's

finding that abortion is an important means of fertility management for women and couples in all states and population subgroups, often as important as, and in some cases more important than contraception, further reinforces the importance of ensuring that women have good access to safe and legal abortion services that meet global standards of care.

The results increase our understanding of the determinants of variations in average family size across states and subpopulations and have implications for meeting the need for sexual and reproductive health services. Health policies and programs in India are generally set at the national level, and the findings of this analysis, that patterns of contributions of the key drivers of fertility are increasingly homogeneous across states and groups reinforces the value of national level policies and programs. At the same time, state governments are largely responsible for implementation, and having evidence on the determinants of family size for each state, and for population subgroups within states, is highly relevant, providing insights into where additional efforts may be needed to ensure that all women are obtaining essential sexual and reproductive health services. This analysis also highlights the need for further in-depth research to understand the specific barriers faced by some sub-groups, especially those that are more vulnerable and disadvantaged in terms of ability to access needed sexual and reproductive services.

## Supporting information

**S1 Table. Selected indicators of sexual and reproductive behaviors and unweighted sample size according to social and economic subgroup: National and by state, India 2015–16.**
(DOCX)

**S2 Table. Estimates of the percent contribution of four key proximate determinants to fertility reduction by age-group, actual ASFR and number of women: By state, India 2015–16.**
(DOCX)

**S3 Table. Estimates of the indices of four key proximate determinants of fertility by age-group: National and by state, India 2015–16.**
(DOCX)

**S4 Table. Estimates of the indices of four key proximate determinants of fertility by sub-group: National and by state, India 2015–16.**
(DOCX)

**S5 Table. Estimates of the indices of four key proximate determinants of fertility, actual TFR, estimated TFR and residual, and estimates of percent contribution of each proximate determinant to fertility reduction: National and by state within TFR groupings, India 2015–16.**
(DOCX)

## Author Contributions

**Conceptualization:** Susheela Singh, Chander Shekhar, Akinrinola Bankole.

**Data curation:** Akinrinola Bankole, Suzette Audam, Temitope Akinade.

**Formal analysis:** Susheela Singh, Chander Shekhar, Akinrinola Bankole.

**Funding acquisition:** Susheela Singh, Chander Shekhar.

**Investigation:** Chander Shekhar.

**Methodology:** Susheela Singh, Chander Shekhar, Akinrinola Bankole, Rajib Acharya.

**Project administration:** Akinrinola Bankole, Suzette Audam, Temitope Akinade.

**Software:** Akinrinola Bankole, Suzette Audam, Temitope Akinade.

**Supervision:** Susheela Singh, Akinrinola Bankole.

**Validation:** Susheela Singh, Chander Shekhar, Akinrinola Bankole, Rajib Acharya, Suzette Audam, Temitope Akinade.

**Visualization:** Chander Shekhar, Akinrinola Bankole, Temitope Akinade.

**Writing – original draft:** Susheela Singh, Chander Shekhar, Akinrinola Bankole, Rajib Acharya, Suzette Audam, Temitope Akinade.

**Writing – review & editing:** Susheela Singh, Chander Shekhar, Akinrinola Bankole, Rajib Acharya, Suzette Audam, Temitope Akinade.

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
