## [Decision Letter · Decision Letter 0]

16 Nov 2021

PONE-D-21-30800Key drivers of fertility levels and differentials in India, at the national, state and population subgroup levels, 2015-2016: An application of Bongaarts’ proximate determinants modelPLOS ONE

Dear Dr. Singh,

Thank you for submitting your manuscript to PLOS ONE. After careful consideration, we feel that it has sufficient  merit but requires minor correction to fully meet PLOS ONE’s publication criteria as it currently stands. Therefore, we invite you to submit a revised version of the manuscript that addresses the points raised during the review process.

Kindly attend to the comments from the reviewer.Also, There are few issues to be attended to:Delete the details about MA, the details add any value to the paragraph unless it has any influence on the abortion estimate used. it is more appropriate to show the sub-national variations of abortion incidence in similar way the other proximate factors were described.(An141 estimated 73 percent of the 15.6 million abortions that took place in 2015 were induced through142 use of MA obtained outside of health facilities (13). However, studies have found that the143 information provided to clients purchasing MA drugs is generally inadequate and inaccurate,144 including essential information on timing of administration of the medication and what side145 effects to expect while experiencing the normal process of the abortion (20,21). Moreover,146 pharmacists themselves have very poor knowledge about the correct regimen of the drugs to147 safely induce an abortion (21,22)):==============================

We look forward to receiving your revised manuscript.

Kind regards,

Akanni Ibukun Akinyemi, PhD

Academic Editor

PLOS ONE

Journal Requirements:

3. Please include a new copy of Tables 1-3, S1 and S2 in your manuscript; the current table is difficult to read. Please follow the link for more information: https://blogs.plos.org/plos/2019/06/looking-good-tips-for-creating-your-plos-figures-graphics/

Additional Editor Comments :

It is a well written manuscript. There are few issues to be attended to:

Delete the details about MA, the details add any value to the paragraph unless it has any influence on the abortion estimate used. it is more appropriate to show the sub-national variations of abortion incidence in similar way the other proximate factors were described.

(An

141 estimated 73 percent of the 15.6 million abortions that took place in 2015 were induced through

142 use of MA obtained outside of health facilities (13). However, studies have found that the

143 information provided to clients purchasing MA drugs is generally inadequate and inaccurate,

144 including essential information on timing of administration of the medication and what side

145 effects to expect while experiencing the normal process of the abortion (20,21). Moreover,

146 pharmacists themselves have very poor knowledge about the correct regimen of the drugs to

147 safely induce an abortion (21,22)):

Part of tables 1 & 2 (last column) was cut off. Please, check the tables.

Table 3 is long. You may consider it as part of the appendix.

Reviewers' comments:

Reviewer's Responses to Questions

**Comments to the Author**

1. Is the manuscript technically sound, and do the data support the conclusions?

Reviewer #1: Yes

2. Has the statistical analysis been performed appropriately and rigorously? 

Reviewer #1: Yes

3. Have the authors made all data underlying the findings in their manuscript fully available?

Reviewer #1: Yes

4. Is the manuscript presented in an intelligible fashion and written in standard English?

Reviewer #1: Yes

5. Review Comments to the Author

Reviewer #1: A well written paper. The focus is of high public health relevance. The paper would be useful in providing evidences that can inform policies and programs that address gaps in access to sexual and reproductive health (SRH) services among Indian states and key population subgroups, through providing evidence on the role of the key determinants of fertility level across states and population groups.

This study was nationally representative, hence its generalizability. The four contributors to fertility levels were explored. Methods were appropriate, results were adequate and well presented. The study limitations were identified and well stated but it appears to have overshadowed the strength of the paper. Can the authors “strengthen” the strength of their studies?

L31: delete: possible because abortion incidence estimates were available for 2015;

The aims and objectives of the paper should have been stated at the end of the introduction. This should be supplied. The aims and objectives is different from the usefulness of the study as currently stated in L97.

The sampling methodology is missing. You should describe it briefly and refer the readers to where it could be found

L304: How reliable/consistent is it to choose the longest duration of any of the four factors—breastfeeding, amenorrhea, postpartum abstinence, and non-contraceptive hysterectomy for each of the women? Rather than choosing a particular one for all the women?

Marriage was explored as one of the determinants, but the actual measurement on marriage was not stated……age at marriage? Type of family? Etc which was considered?

L357 – 361 - 371: Move to methods

Table 1 : The last columns were cut off

Table 2, 3 : The last columns were cut off

Table 3 is too long…could it be presented as for example Table 3a for a specific state/region….else…Table 3 may go for a supplementary material

There are no specific recommendations for policymakers and population programmers in India. This must be provided.

6. PLOS authors have the option to publish the peer review history of their article (what does this mean?). If published, this will include your full peer review and any attached files.

Reviewer #1: **Yes: **Adeniyi Francis Fagbamigbe

---

## [Author Response · Author response to Decision Letter 0]

17 Dec 2021

Manuscript Content

• Delete the details about MA, the details add any value to the paragraph unless it has any influence on the abortion estimate used. it is more appropriate to show the sub-national variations of abortion incidence in similar way the other proximate factors were described.

(An

141 estimated 73 percent of the 15.6 million abortions that took place in 2015 were induced through

142 use of MA obtained outside of health facilities (13). However, studies have found that the

143 information provided to clients purchasing MA drugs is generally inadequate and inaccurate,

144 including essential information on timing of administration of the medication and what side

145 effects to expect while experiencing the normal process of the abortion (20,21). Moreover,

146 pharmacists themselves have very poor knowledge about the correct regimen of the drugs to

147 safely induce an abortion (21,22))

Response: We agree that specifics on MA are not relevant for this analysis and have deleted those details. We have added information on what data on abortion incidence is available, and therefore used for the current analysis. 

• The study limitations were identified and well stated but it appears to have overshadowed the strength of the paper. Can the authors “strengthen” the strength of their studies?

Response: We agree and have re-organized parts of the Discussion and edited the text to highlight the contributions of the study, while reducing the over-emphasis on limitations (by reducing repetition, focusing on study limitations as opposed to data limitations, and referencing future research needs to recognize areas where data are inadequate). 

• L31: delete: possible because abortion incidence estimates were available for 2015;

Response: Agree and deletion made.

• The aims and objectives of the paper should have been stated at the end of the introduction. This should be supplied. The aims and objectives is different from the usefulness of the study as currently stated in L97.

Response: Agree and added to the text to address this suggestion. 

• The sampling methodology is missing. You should describe it briefly and refer the readers to where it could be found.

Response: We have added a sentence on sampling methodology and study design for the two main secondary data sources used in this analysis (NFHS4 and a study on abortion incidence), and also provided references to published documentation on the methodology underlying these data sources.

• L304: How reliable/consistent is it to choose the longest duration of any of the four factors—breastfeeding, amenorrhea, postpartum abstinence, and non-contraceptive hysterectomy for each of the women? Rather than choosing a particular one for all the women?

Response: This is a standard demographic approach to integrating the impact of two or more behaviors that contribute to an outcome. This is logical because each behavior contributes to reduction in exposure to pregnancy and because individual women vary in practice of each behavior. In the Demographic and Health Surveys, for which most countries collect data on the first three of these four behaviors, the standard measure of infecundability is based on the behavior with the longest duration. Our adaptation to the standard measure is to add in one more behavior that is especially relevant in India, that is non-contraceptive hysterectomy.

• Marriage was explored as one of the determinants, but the actual measurement on marriage was not stated……age at marriage? Type of family? Etc which was considered?

Response: The index of marriage is based on proportions currently married for five-year age groups; it is a proxy for sexual activity and likelihood of pregnancy. We have added this information to the text in the methods section. This index does not incorporate type of relationship or family structure. 

• L357 – 361 - 371: Move to methods

Response: We agree with the Reviewer that this paragraph fits better in the Methods section and have moved it to be the last paragraph of the methods section. This has the advantage that the reader will see it just before moving into Results, where those points are relevant and necessary for understanding the results that are reported. 

• Table 3 is too long…could it be presented as for example Table 3a for a specific state/region….else…Table 3 may go for a supplementary material. 

Response: We agree and have responded by dividing Table 3 into four segments, numbered Tables 3a, 3b, 3c and 3d. Each of these tables presents results for one socio-economic characteristic at the national level and for each state, and the four socioeconomic characteristics are: place of residence, educational attainment, wealth status, and caste. This approach has two important advantages -- in addition to having tables that are easier to digest, it facilitates comparisons across states for each characteristic. Our focus in interpretation of results is comparisons across states for each characteristic and not on within state comparisons across characteristics: Therefore, this change in presentation of results better serves interpretation of the results. We prefer to keep these tables within the body of the paper (rather than presenting them as an Appendix Table) because results from the subgroup analysis are a key contribution of the paper. 

• There are no specific recommendations for policymakers and population programmers in India. This must be provided.

Response: We agree that more specific recommendations are needed. We have added a section on recommendations for policymakers and providers, particularly with respect to improvements needed in contraceptive services and abortion services. We also offer some recommendation for improving key relevant data. 

Formatting and Submission

Authors Response regarding “Data not Shown” : We have added Appendix Table 5 that presents findings according to groups of states by TFR level, and we reference this appendix table in the text. 

3. Please include a new copy of Tables 1-3, S1 and S2 in your manuscript; the current table is difficult to read. Please follow the link for more information: https://blogs.plos.org/plos/2019/06/looking-good-tips-for-creating-your-plos-figures-graphics/

 Authors Response: We have uploaded new copies of all tables, figures and supplementary Appendix Tables.

Authors Response: We have checked references; we do not cite any retracted articles or other retracted sources.

---

## [Editor Report · Decision Letter 1]

21 Jan 2022

Key drivers of fertility levels and differentials in India, at the national, state and population subgroup levels, 2015-2016: An application of Bongaarts’ proximate determinants model

PONE-D-21-30800R1

Dear Dr. Singh,

We’re pleased to inform you that your manuscript has been judged scientifically suitable for publication and will be formally accepted for publication once it meets all outstanding technical requirements.

Kind regards,

Akanni Ibukun Akinyemi, PhD

Academic Editor

PLOS ONE
---

## [Editor Report · Acceptance letter]

28 Jan 2022

PONE-D-21-30800R1 

Key drivers of fertility levels and differentials in India, at the national, state and population subgroup levels, 2015-2016: An application of Bongaarts’ proximate determinants model 

Dear Dr. Singh:

I'm pleased to inform you that your manuscript has been deemed suitable for publication in PLOS ONE. Congratulations! Your manuscript is now with our production department. 

Kind regards, 

on behalf of

Dr. Akanni Ibukun Akinyemi 

Academic Editor

PLOS ONE